# Filamin A organizes γ-aminobutyric acid type B receptors at the plasma membrane

Marie-Lise Jobin [1,2,9] ✉, Sana Siddig [1,2], Zsombor Koszegi [3,4], Yann Lanoiselée[3,4], Vladimir Khayenko [5], Titiwat Sungkaworn[1,2], Christian Werner [6], Kerstin Seier [1,2], Christin Misigaiski[1,2], Giovanna Mantovani[7,8], Markus Sauer [6], Hans M. Maric [5] & Davide Calebiro [1,2,3,4] ✉

The γ-aminobutyric acid type B (GABA_B) receptor is a prototypical family C G protein-coupled receptor (GPCR) that plays a key role in the regulation of synaptic transmission. Although growing evidence suggests that GPCR signaling in neurons might be highly organized in time and space, limited information is available about the mechanisms controlling the nanoscale organization of GABA_B receptors and other GPCRs on the neuronal plasma membrane. Using a combination of biochemical assays in vitro, single-particle tracking, and super-resolution microscopy, we provide evidence that the spatial organization and diffusion of GABA_B receptors on the plasma membrane are governed by dynamic interactions with filamin A, which tethers the receptors to sub-cortical actin filaments. We further show that GABA_B receptors are located together with filamin A in small nanodomains in hippocampal neurons. These interactions are mediated by the first intracellular loop of the GABA_B1 subunit and modulate the kinetics of Gα_i protein activation in response to GABA stimulation.

The γ-aminobutyric acid type B (GABA_B) receptor is the metabotropic receptor for the main inhibitory neurotransmitter in the central nervous system and, as such, plays a key role in the modulation of neuronal activity[1]. GABA_B receptor signaling is involved in several physiological processes, such as locomotion and nociception[2]. Moreover, alterations of GABA_B receptor activity have been implicated in the etiology of conditions such as drug addiction and epilepsy[3], making GABA_B receptors promising therapeutic targets[4,5].

GABA_B is a family C G protein-coupled receptor (GPCR) consisting of an obligatory heterodimer of a GABA_B1 and a GABA_B2 subunit. The GABA_B1 subunit bears the ligand binding site, whereas the GABA_B2 subunit binds and activates G_i/o proteins composed of a Gα_i/o subunit and a Gβγ dimer[6–9]. The Gα_i/o subunit released upon GABA_B activation inhibits adenylyl cyclase activity, resulting in a reduction of intracellular cAMP levels, whereas the Gβγ dimer inhibits presynaptic voltage-dependent Ca²⁺ channels (VDCCs) and activates postsynaptic G protein-activated inwardly rectifying potassium (GIRK) channels[10].

The heterodimerization of GABA_B1 and GABA_B2 has been shown to be mediated by coiled-coil interactions between two homologous α-helical domains present in the intracellular C-tails of both GABA_B1 and GABA_B2 subunits[11–13], as well as between their two extracellular Venus flytrap domains[14] and, as more recently shown, through an

[1]Institute for Pharmacology and Toxicology, University of Würzburg, Würzburg, Germany. [2]Bio-Imaging Center/Rudolf Virchow Center, University of Würzburg, Würzburg, Germany. [3]Institute of Metabolism and Systems Research, University of Birmingham, Birmingham, UK. [4]Centre of Membrane Proteins and Receptors (COMPARE), Universities of Birmingham and Nottingham, Nottingham, UK. [5]Rudolf Virchow Center, Center for Integrative and Translational Bioimaging, University of Würzburg, Würzburg, Germany. [6]Department of Biotechnology and Biophysics, Biocenter, University of Würzburg, Würzburg, Germany. [7]Endocrinology Unit, Fondazione IRCCS Ca' Granda Ospedale Maggiore Policlinico, Milan, Italy. [8]Department of Clinical Sciences and Community Health, University of Milan, Milan, Italy. [9]Present address: Interdisciplinary Institute for Neuroscience (IINS), CNRS UMR5297, University of Bordeaux, 33000 Bordeaux, France. ✉e-mail: marie-lise.jobin@u-bordeaux.fr; D.Calebiro@bham.ac.uk

intersubunit "latch" involving the transmembrane domains[15,16]. Importantly, the coiled-coil interaction between the two C-tails masks an endoplasmic reticulum (ER) retention signal in the GABA$_{B1}$ subunit, thus ensuring that only functional GABA$_B$ heterodimers reach the cell surface[11]. Moreover, GABA$_B$ receptors can form higher-order oligomers, the stability of which has been shown to be modulated by receptor activation[17–19].

GABA$_B$ receptors are located both pre- and postsynaptically in the central nervous system[8,20]. Presynaptic GABA$_B$ receptors negatively modulate neurotransmitter release via the G$\alpha_{i/o}$-mediated inhibition of adenylyl cyclases and G$\beta\gamma$-mediated inhibition of VDCCs and the resulting reduction of Ca$^{2+}$ entry, which is required for vesicle fusion[2,21]. In addition, the G$\beta\gamma$ dimer binds the soluble N-ethylmaleimide-sensitive factor attachment protein receptor (SNARE) complex required for vesicle fusion, thereby limiting neurotransmitter release downstream of Ca$^{2+}$ entry[2,10]. Postsynaptic GABA$_B$ receptors decrease neuronal excitability by inducing membrane hyperpolarization mediated by the G$\beta\gamma$-dependent stimulation of GIRK channel currents[8].

GABA$_B$ receptors are expressed in almost all regions of the brain, yet they show high functional diversity, which suggests the possible existence of specific modulators of their plasma membrane localization and/or function[20,22]. In a previous study, we suggested that GABA$_B$ receptors might undergo dynamic interactions with the actin cytoskeleton, which might control their nanoscale location within the plasma membrane[18]. However, the molecular mechanisms and the functional consequences of these interactions remain enigmatic. Here, we use a combination of single-molecule microscopy, real-time fluorescence resonance energy transfer (FRET) measurements and in vitro assays to elucidate these key mechanisms at the basis of GABA$_B$ receptor spatiotemporal organization and function.

## Results

### GABA$_B$ receptors colocalize transiently with the cytoskeleton

Our previous work revealed that GABA$_B$ receptors partially arrange in rows on the plasma membrane following the underlying actin fibers[18], suggesting that GABA$_B$ receptors either transiently interact with the actin cytoskeleton or that different populations of receptors might coexist. To further investigate these hypotheses, we imaged fluorescently labelled GABA$_B$ receptors at the surface of Chinese hamster ovary (CHO) cells by total internal reflection fluorescence (TIRF) microscopy. CHO cells were initially co-transfected for 24 hours (h) with a GABA$_{B1}$ construct bearing an N-terminal SNAP-tag[23], untagged GABA$_{B2}$ and Lifeact-GFP to label actin fibers (Fig. 1a). The SNAP-tagged GABA$_{B1}$ was labelled by incubation with an irreversible, membrane impermeable fluorescent SNAP substrate (SNAP-549). Representative TIRF images obtained with the different construct combinations used in this study are shown in Supplementary Fig. 1. We found GABA$_B$ receptors to be preferentially arranged in rows on the plasma membrane, apparently following the underlying actin fibers (Fig. 1b and Supplementary Fig. 1a, b), consistent with our previous observations. A quantitative analysis based on Manders' colocalization coefficients (MCCs)[24], revealed a high degree of colocalization between GABA$_B$ and actin fibers (MCC 0.77; Fig. 1c). Note that MCCs do not necessarily give a value of 0 even in the case when one of the two channels has a perfect homogenous distribution. For comparison, CD86, an unrelated membrane protein that has no relevant interactions with GPCRs, G proteins or the actin cytoskeleton and was used as control[18,25], gave a MCC of 0.34 (Fig. 1c, d and Supplementary Fig. 1c), whereas replacing the GABA$_B$ images with synthetic homogenous distributions, which demonstrates the lowest theoretically possible MCC values, gave a MCC of 0.15 (Fig. 1c). We then reduced the transfection time to ~4 h, which allowed us to visualize individual GABA$_B$ receptors diffusing on the surface of living cells (Supplementary Movie 1). This revealed the presence of individual receptors alternating phases of free diffusion and transient immobilization along actin fibers, consistent with the

underlying interactions being dynamic (Fig. 1e, Supplementary Movies 2, 3). Applying our previously developed single-molecule colocalization analysis[25] we obtained a colocalization index of $0.09 \pm 0.01$ (Fig. 1f), which is significantly higher than expected by chance (see comparison with homogenous distribution) or measured with CD86. These results indicated that the interaction of GABA$_B$ receptors with the actin cytoskeleton occurs also at low/physiological receptor expression levels and is dynamic.

### Filamin A tethers GABA$_B$ receptors to the actin cytoskeleton

Since interactions of membrane proteins with the actin cytoskeleton are usually mediated by scaffold proteins, we next sought to identify a potential scaffold tethering GABA$_B$ receptors to actin fibers. Among a few possible candidates, we focused our attention on filamin A (FLNA), which previous studies by other groups and ours have implicated in the anchoring of other GPCRs to the actin cytoskeleton[26–28].

First, we co-expressed eGFP-tagged FLNA with SNAP-GABA$_{B1}$, which was labeled with SNAP-647, and GABA$_{B2}$ in CHO cells (Fig. 2a). Similar to actin, a high colocalization between GABA$_B$ and FLNA was present (MCC 0.87) compared with what was observed in the case of CD86 (MCC 0.44) or synthetic homogenous distributions (MCC 0.15) (Fig. 2a, b). To test the involvement of FLNA in the organization of GABA$_B$ at the plasma membrane, we co-expressed a dominant-negative FLNA fragment corresponding to repeats 19 and 20 (FLNA19-20) fused to the fluorescent protein DsRed for visualization (Fig. 2c), which was previously successfully used to disrupt the interactions of FLNA with the somatostatin receptor 2 (SST2) in vitro and in living cells[26,28,29]. This fragment contains two of the FLNA repeats that are frequently described to be responsible for binding to membrane proteins but lacks the domains mediating actin binding. As a result, FLNA19-20 competes with endogenous FLNA for binding to membrane proteins, thus interfering with their FLNA-mediated actin tethering without altering the overall organization of the actin cytoskeleton[26,28,29]. A FLNA fragment encompassing repeats 17 and 18 (FLNA17-18) (Fig. 2c), which does not interact with SST2 or other tested GPCRs, was used as a negative control as done in previous studies[26,28,29]. Notably, overexpression of FLNA19-20 caused a statistically significant reduction of GABA$_B$ receptor colocalization with F-actin compared with the negative control FLNA17-18 (Fig. 2d, e). To further confirm the impact of FLNA on GABA$_B$ plasma membrane organization, we performed siRNA silencing of FLNA in CHO cells, which caused a clear loss of the typical GABA$_B$ striated pattern (Supplementary Fig. 2). As a note, we failed to generate either CHO or human embryonic kidney 293 (HEK293) cells lacking FLNA expression by CRISPR/Cas9 gene editing, which prevented their use in this study. Overall, these results indicate that FLNA tethers GABA$_B$ receptors to the subcortical actin cytoskeleton.

Next, we explored the role of FLNA in GABA$_B$ receptor localization in neurons. Primary mouse hippocampal neurons were labeled with a GABA$_{B1}$ antibody together with either fluorescently conjugated phalloidin to stain F-actin (Fig. 2f) or a FLNA antibody (Fig. 2g). The results revealed a partial but robust colocalization between GABA$_B$ and both actin fibers (Fig. 2h) and FLNA (Fig. 2i) (MCC 0.39 and 0.31, respectively). The colocalization involved small diffraction-limited domains apparently located on the cell body and along fine neuronal protrusions. To further resolve the organization of GABA$_B$ and FLNA in those domains, we applied super-resolution 4X expansion microscopy, achieving a lateral resolution of 60-70 nm[30]. Confocal imaging of the expanded hippocampal neurons consistently showed partial colocalization between GABA$_B$ and FLNA in small nanodomains prevalently located on dendritic shafts and fine neuronal protrusions (Fig. 2j, arrowheads). For comparison, we simultaneously stained hippocampal neurons with two distinct antibodies both recognizing the presynaptic active zone marker Bassoon, a condition where maximal colocalization is expected (Supplementary Fig. 3). Only slightly higher colocalization

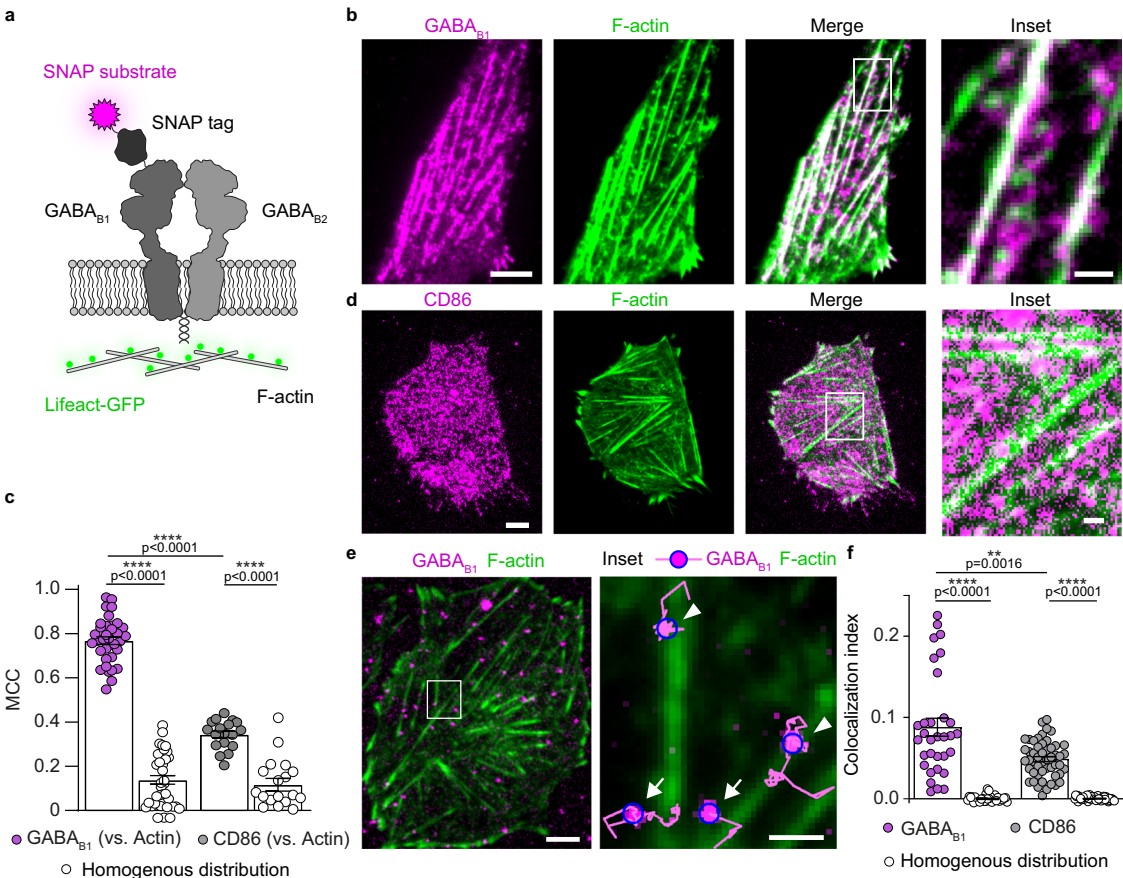

**Fig. 1 | GABA_B receptors are transiently tethered along actin fibers. a** Scheme of the constructs used in the study. The GABA_{B1} construct contains a SNAP-tag at its N-terminus, allowing direct labeling with the fluorescent benzylguanine derivative SNAP-Surface 549 (SNAP-549). **b** Representative TIRF images of CHO cells co-transfected with SNAP-GABA_{B1} (magenta), GABA_{B2}, and Lifeact-GFP (green) for 24 h. The inset corresponds to the region delimited by the white box.
**c** Quantification of GABA_B and CD86 colocalization with F-actin. Shown are Manders' colocalization coefficient (MCC) values compared with those expected for homogenous GABA_B distributions or observed with CD86. The bars show mean ± SEM values. n = 35 cells and 18 cells (GABA_B and CD86, respectively) examined over three independent experiments. ****$p < 0.0001$ by two-tailed unpaired Welch's t-test. **d** Representative TIRF images of CHO cells co-transfected with SNAP-CD86

(magenta) and Lifeact-GFP (green) for 24 h. The inset corresponds to the region delimited by the white box. **e** Representative frame of a single-molecule movie of SNAP-GABA_{B1} labeled with SNAP-549 (magenta) in a cell additionally co-transfected with GABA_{B2} and Lifeact-GFP (green). The inset corresponds to the region delimited by the white box. In the inset, each detected particle is surrounded by a blue circle and particle trajectories are shown in magenta. Arrowheads, receptors transiently trapped on actin fibers. Arrows, receptors diffusing between actin fibers. **f** Single-molecule colocalization indexes calculated from images like in **e**. The bars show mean ± SEM. n = 32 and 51 cells (GABA_B and CD86, respectively) examined over three independent experiments. **$p < 0.01$, ****$p < 0.0001$ by two-tailed unpaired Welch's t-test. Scale bars, 5 μm (images), 1 μm (insets). Source data are provided as a Source Data file.

coefficients were measured between the two Bassoon stainings than between GABA_B and FLNA (MCC 0.57 vs. 0.36, respectively), which was not statistically significant (Fig. 2k). These results indicated that a relevant fraction of GABA_B is in close proximity to FLNA within diffraction-limited nanodomains in hippocampal neurons.

**FLNA binds the first intracellular loop of GABA_{B1}**
We then set out to identify the GABA_B region that mediates actin anchoring via FLNA. When SNAP-tagged GABA_{B2} was transfected alone (Supplementary Fig. 1b), no striated pattern was observed at the plasma membrane. Moreover, when the ER retention motif in the C-tail of the GABA_{B1} subunit was deleted to allow it to reach the plasma membrane in the absence of GABA_{B2} co-transfection, the GABA_{B1} subunit still showed a striated pattern at the cell surface (Supplementary Fig. 1d). Altogether, these results indicated that the domain responsible for interaction with FLNA is residing within the GABA_{B1} subunit.

   To further restrict the GABA_{B1} region responsible for actin anchoring, we first generated GABA_{B1} mutants in which the C-tail was progressively truncated (Supplementary Fig. 1e–h). No changes were

observed in the organization of GABA_{B1} at the surface of CHO cells caused by either partial or complete removal of the C-tail, ruling out an involvement of this region. Next, we replaced each of the three GABA_{B1} intracellular loops with the corresponding ones in GABA_{B2} (Supplementary Fig. 1i–k). Replacing either intracellular loop also did not cause any visible changes in the GABA_B pattern at the cell surface, possibly due to partial sequence homology between GABA_{B1} and GABA_{B2}. We, therefore, decided to replace the GABA_{B1} loops with those of the more distantly related metabotropic glutamate receptor 2 (mGluR2), which, similar to GABA_{B2}, does not show any preferential colocalization with the actin cytoskeleton (Supplementary Fig. 1l). Remarkably, replacing all three intracellular loops with those of mGluR2 abolished GABA_B striated pattern (Supplementary Fig. 1m). Similar results were obtained when only the first intracellular loop (IL1) was replaced (Supplementary Fig. 1n), whereas replacing either the second or third intracellular loop had no effect (Supplementary Fig. 1o, p). These findings were further supported by experiments in which we measured the colocalizations of GABA_B with either F-actin (Fig. 3a) or FLNA (Fig. 3b), which were both largely reduced by replacement of GABA_B IL1 with that of mGluR2. These results

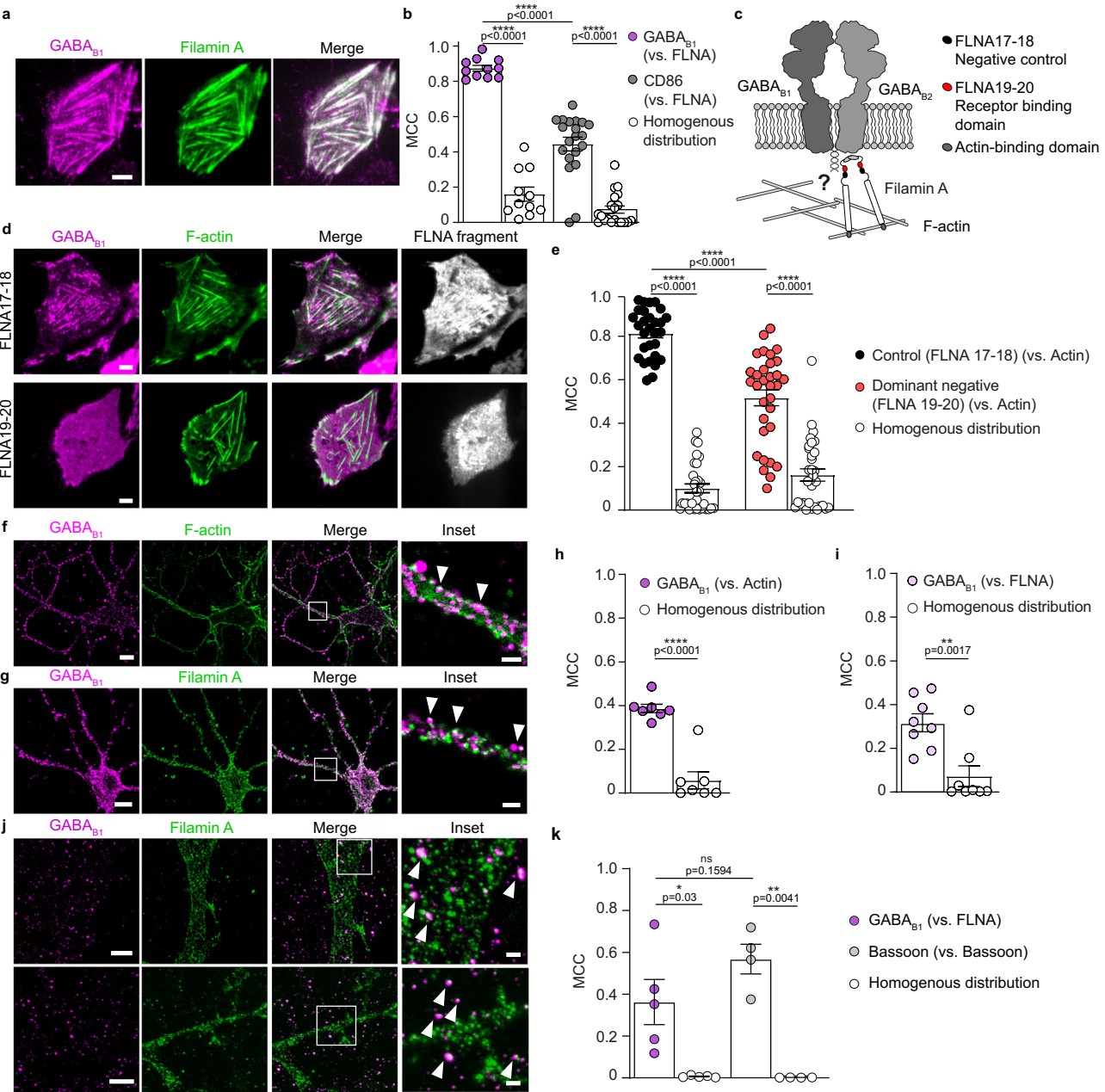

**Fig. 2 | FLNA mediates GABA_B receptor association with the actin cytoskeleton.**
**a** Representative TIRF images of CHO cells transfected with SNAP-GABA_{B1} (magenta), GABA_{B2} and eGFP-FLNA (green) for 24 h and labeled with SNAP-647. White color in the merge image is indicative of colocalization. **b** Quantification of GABA_B and CD86 colocalization with FLNA. Shown are MCC values compared with those expected for homogenous distributions. Data are mean ± SEM. $n = 11$ and 20 cells (GABA_B and CD86, respectively) examined over two independent experiments. **c** Scheme of the FLNA dimer with each subunit composed of an N-terminal actin-binding domain attached to 24 immunoglobulin-like domains. **d** Representative TIRF images of CHO cells co-transfected with SNAP-GABA_{B1} and Lifeact-GFP and either DsRed-FLNA17-18 or DsRed-FLNA19-20. **e** Quantification of the effect of FLNA19-20 on the colocalization between SNAP-GABA_{B1} and Lifeact-GFP. Shown are MCC values vs. those of homogenous GABA_{B1} distributions. Data are mean ± SEM. $n = 29$ and 32 cells (FLNA17-18 and FLNA19-20, respectively) examined over three independent experiments. **f, g** Representative confocal microscopy images of hippocampal neurons immunostained for GABA_{B1} (magenta) and either actin (green) (**f**) or FLNA (green) (**g**). Insets correspond to the regions delimited by the white boxes. Arrowheads, examples of colocalization. **h, i** Quantification of GABA_{B1} colocalization with actin (**h**) or FLNA (**i**) in hippocampal neurons. Shown are MCC values vs. those of homogenous GABA_{B1} distributions. Data are mean ± SEM. $n = 7$ and 8 cells (actin and FLNA, respectively) examined over three independent experiments. **j** Maximum-intensity projection of confocal microscopy stacks after 4X expansion microscopy of hippocampal neurons immunostained for GABA_{B1} (magenta) and FLNA (green). **k** Quantification of GABA_{B1} colocalization with FLNA based on 4X expansion microscopy images. Shown are MCC values vs. those of homogenous GABA_{B1} distributions. Data are mean ± SEM. $n = 5$ and 4 cells (GABA_{B1} and Bassoon, respectively) examined over two independent experiments. \*$p < 0.05$, \*\*$p < 0.01$, \*\*\*\*$p < 0.0001$ by two-tailed unpaired Welch's t-test. ns statistically not significant. Scale bars, 5 μm (images in **a**, **d**), 10 μm (images in **f**, **g**, **j**), 2 μm (insets in **f**, **g**, **j**). Source data are provided as a Source Data file.

indicated that the IL1 of GABA_{B1} is involved in the anchoring of GABA_B to the actin cytoskeleton.

To define the exact GABA_B binding site recognized by FLNA, we synthesized the individual intracellular loops as well as the C-tails of both GABA_{B1} and GABA_{B2} in the form of an overlapping 15-mer peptide library. Peptide microarrays containing this library were then used to probe the GABA_B regions capable of binding FLNA19-20 in vitro. FLNA17-18 was used as negative control (Supplementary Fig. 4).

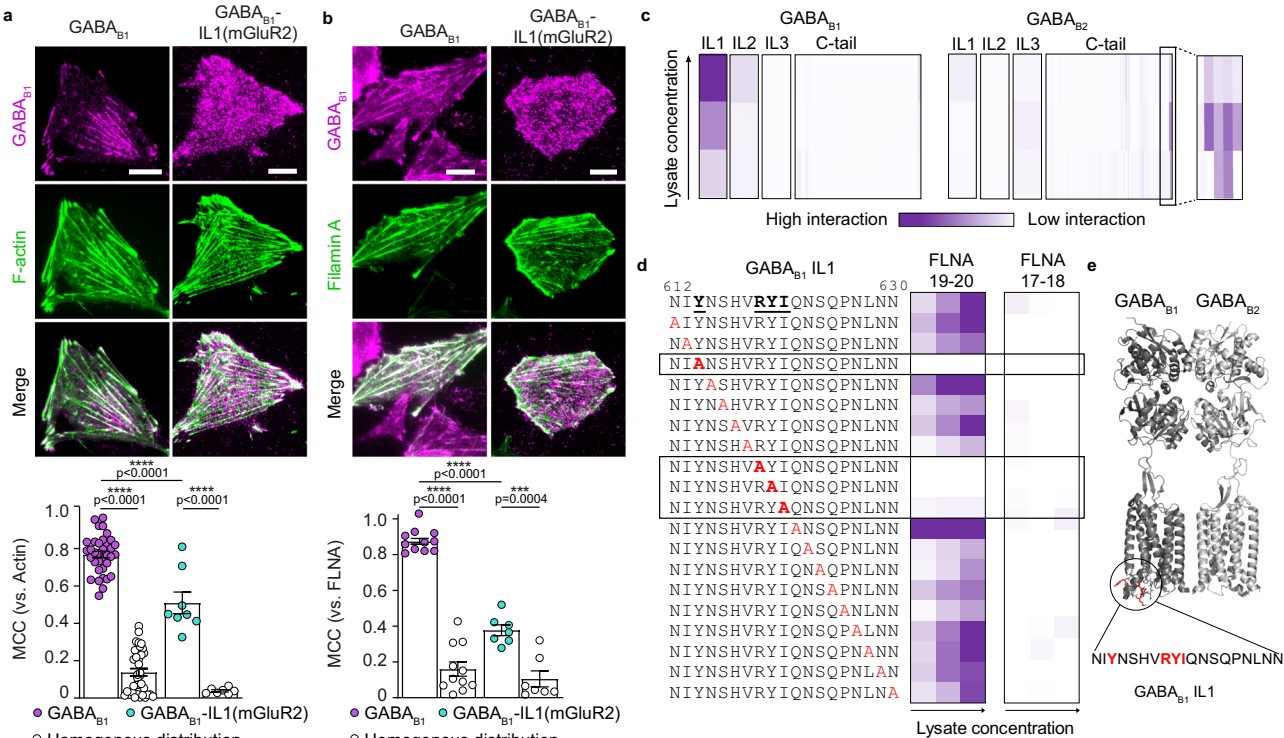

**Fig. 3 | A linear motif in the IL1 of GABA_B1 binds FLNA to mediate interaction with the actin cytoskeleton. a** Top, representative TIRF images of SNAP-GABA_B1 (left) or SNAP-GABA_B1-IL1(mGluR2) (right) labeled with SNAP-647 in CHO cells additionally co-transfected with GABA_B2 and Lifeact-GFP. Bottom, corresponding MCC analyses. Data are mean ± SEM. $n = 35$ and $n = 8$ (GABA_B1 and GABA_B1-IL1(m-GluR2), respectively) examined over three independent experiments. **b** Top, representative TIRF images of SNAP-GABA_B1 (left) or SNAP-GABA_B1-IL1(mGluR2) (right) labeled with SNAP-647 in CHO cells co-transfected with GABA_B2 and eGFP-FLNA. Bottom, corresponding MCC analyses. Data are mean ± SEM. $n = 11$ and $n = 7$ (GABA_B1 and GABA_B1-IL1(mGluR2), respectively) examined over two independent experiments. **c**, Peptide microarray results. Peptide microarrays encompassing GABA_B1 and GABA_B2 intracellular domains were incubated with increasing concentrations of cell lysates containing DsRed-FLNA19-20. The fluorescence readout is plotted as a heatmap, normalized to the strongest detected signal. A linear dependency between signal intensity and lysate concentration was observed. **d** Screening with a microarray containing single-point alanine mutations of the GABA_B1 IL1. Data are presented as in **c**. **e** Cartoon representation of GABA_B inactive structure (PDB ID: 6WIV)[15] highlighting the IL1 as sticks and the amino acids involved in the FLNA interaction in red. Experiments in **c** and **d** were performed twice for each condition, each time in duplicate. ***$p < 0.001$, ****$p < 0.0001$ by two-tailed unpaired Welch's *t*-test. Scale bars, 10 μm. Source data are provided as a Source Data file.

The results revealed a specific binding of FLNA19-20 to peptides corresponding to the IL1 of GABA_B1 (Fig. 3c). A second potential interaction of lower strength was detected with the 4 C-terminal amino acids of GABA_B2 (VSGL), a PDZ-binding motif known to interact with several intracellular proteins[31,32]. No interactions were detected with the C-tail of GABA_B1 or any other intracellular regions of either receptor subunit (Fig. 3c). To identify the individual residues in the IL1 of GABA_B1 involved in FLNA binding, we utilized an additional peptide microarray containing peptides corresponding to the IL1 of GABA_B1 in which we performed a single-point alanine scan (Fig. 3d). Substitution of either Tyr(Y)_614, Arg(R)_619, Tyr(Y)_620 or Ile(I)_621 abolished the interaction with FLNA19-20. These results reveal a previously unknown motif in the IL1 of GABA_B1 mediating FLNA binding (Fig. 3e).

**FLNA dynamically modulates GABA_B cell surface mobility**

Having identified a new interaction between GABA_B and the actin cytoskeleton mediated by FLNA, we moved to test whether FLNA binding impacts the lateral diffusion of GABA_B receptors on the plasma membrane. For this purpose, we performed single particle tracking experiments of individual SNAP-tagged GABA_B1 molecules labeled with SNAP-647 in CHO cells co-transfected with GABA_B2 and DsRed-tagged FLNA fragments (Fig. 4a, b). A time-averaged mean square displacement (TAMSD) analysis[33] revealed a large heterogeneity in the diffusivity of individual GABA_B receptors on the plasma membrane (Supplementary Fig. 5a). The TAMSD results were then used to classify each receptor particle in four categories corresponding to virtually immobile or sub- (i.e., confined), free (i.e., normal) and super- (i.e., directional) diffusion (Supplementary Fig. 5a). The presence of the FLNA dominant-negative fragment (FLNA19-20) caused a modest decrease in the relative fraction of immobile receptors (Supplementary Fig. 5b) compared to cells expressing the control fragment (FLNA17-18). This was accompanied by statistically significant increases in the average diffusion coefficients ($D$) of the particles in both the normal and super-diffusion groups, consistent with a role of FLNA in restricting the lateral diffusion of GABA_B receptors on the plasma membrane (Fig. 4c).

To further investigate the spatio-temporal dynamics of the interactions between GABA_B and FLNA, we acquired fast, two-color single-molecule image sequences in CHO cells co-expressing SNAP-GABA_B1, GABA_B2 and CLIP-FLNA (Fig. 4d). Automated single-particle tracking allowed us to detect colocalization events between individual GABA_B and FLNA molecules and estimate both their relative frequency and duration (Fig. 4e, f and Supplementary Movie 4). Average GABA_B-FLNA colocalization times were in the range of 0.2–0.3 s, consistent with the occurrence of transient interactions. Upon GABA stimulation (100 μM; 5 min incubation), we observed a reduction in the relative frequency of GABA_B-FLNA colocalization events to values observed with the GABA_B1-IL1(mGluR2) chimera, used as a non-interacting control (Fig. 4e). This was accompanied by a modest reduction in the average duration of the colocalization events (Fig. 4f).

Furthermore, we used a recently developed statistical approach[34] to segment GABA_B and FLNA trajectories into smaller ones

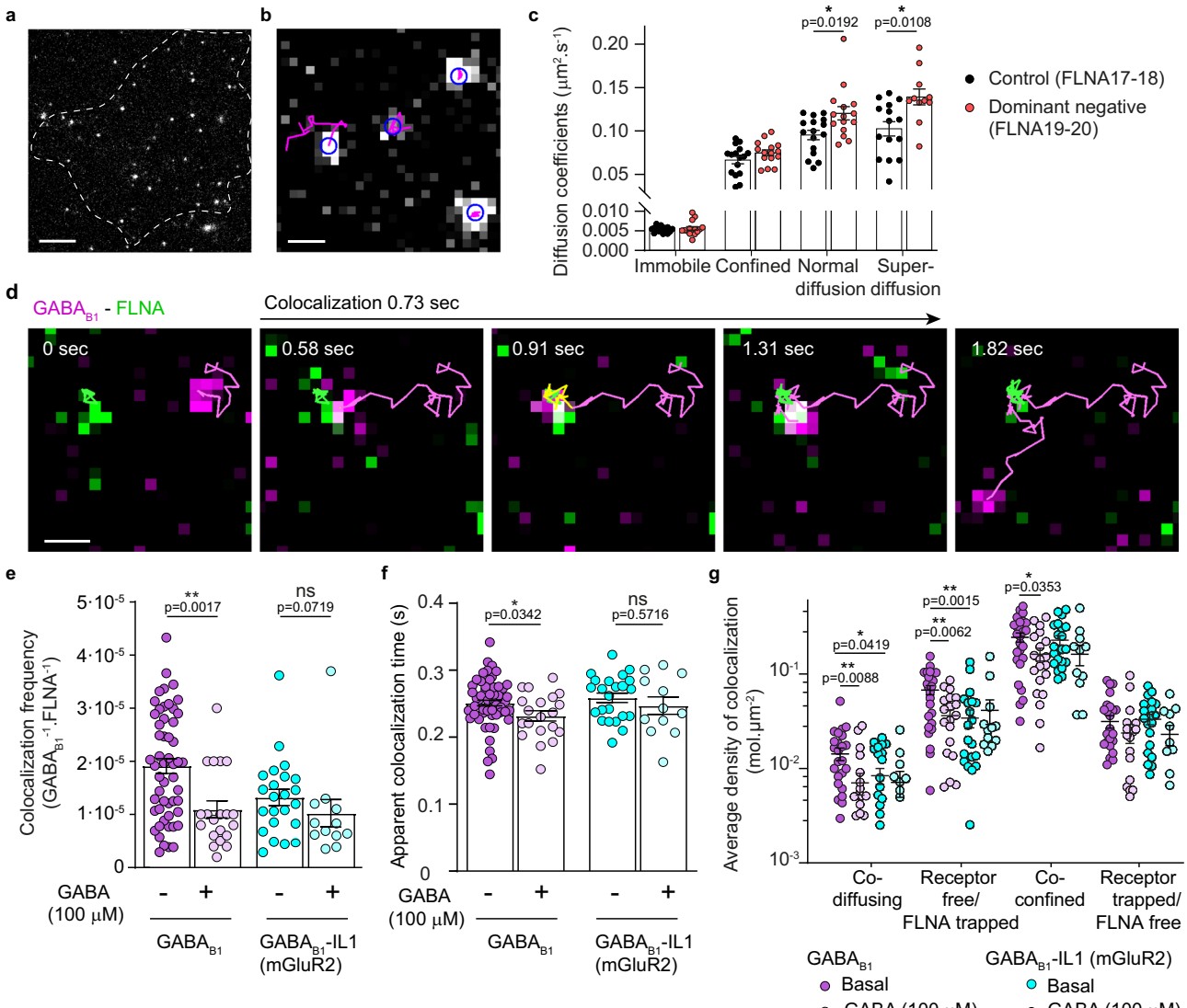

**Fig. 4 | FLNA modulates GABA$_B$ receptor diffusion at the cell surface.**
**a** Representative frame of a single-molecule TIRF image sequence of SNAP-GABA$_{B1}$ labeled with SNAP-647 in CHO cells co-transfected with GABA$_{B2}$ and DsRed-FLNA19-20 or DsRed-FLNA17-18 as negative control. Images are representative of at least three independent experiments. **b** Representative outcome of single-particle tracking from at least three independent experiments. Each detected particle is surrounded by a blue circle and particle trajectories are shown in magenta. **c** Diffusion coefficients of GABA$_B$ receptor particles in the four groups identified by the TAMSD analysis. Data are mean ± SEM from $n = 16$ and 15 cells (2809 and 2670 trajectories) for FLNA17-18 and FLNA19-20, respectively, examined over three independent experiments. **d** Single-molecule analysis of GABA$_B$-FLNA

interactions. CHO cells were co-transfected with SNAP-GABA$_{B1}$ and CLIP-FLNA and labeled with SNAP-647 and CLIP-TMR, respectively. A representative example of a transient colocalization event between a GABA$_B$ and a FLNA molecule is shown. **e–g** Relative frequency (**e**), duration (**f**), and density of diffusivity states (**g**) of single-molecule colocalizations between FLNA and either GABA$_{B1}$ or GABA$_{B1}$-IL1(mGluR2) under basal and stimulated conditions (GABA 100 μM; 5 min incubation). Data are mean ± SEM. $n = 53$ (GABA$_{B1}$ basal), 20 (GABA$_{B1}$ stimulated), 23 (GABA$_{B1}$-IL1(mGluR2) basal) and 12 (GABA$_{B1}$-IL1(mGluR2) stimulated) examined over three independent experiments. $*p < 0.05$, $**p < 0.01$ by two-tailed Mann-Whitney U test. ns statistically not significant. Scale bars, 5 μm (**a**), 500 nm (**b**, **d**). Source data are provided as a Source Data file.

corresponding to phases of free diffusion and confinement, which allowed us to better evaluate the diffusion states of GABA$_B$ and FLNA when they colocalize (see "Trapping analysis" in the Methods section)[34]. Different sub-populations of colocalizing GABA$_B$ and FLNA were observed (Fig. 4g): a co-diffusing one in which they colocalize while both diffusing; a co-confined one where GABA$_B$ and FLNA colocalize while being both trapped; a sub-population of freely diffusing GABA$_B$ that transiently colocalize with trapped FLNA, and a subpopulation of trapped GABA$_B$ that transiently colocalize with freely diffusing FLNA. Importantly, we observed that GABA$_B$ and FLNA are mainly co-confined when they colocalize (Fig. 4g). A smaller proportion of GABA$_B$ (or FLNA) particles are freely diffusing while they encounter a trapped FLNA (or GABA$_B$) particle (Fig. 4g). Upon GABA

stimulation, we observed a variable decrease in all colocalizing subpopulations, which was not detected in the case of GABA$_B$-IL1 (mGluR2), used as control (Fig. 4g). Overall, these results suggest that FLNA and GABA$_B$ are prevalently trapped during their transient colocalizations, which typically occur on actin fibers and are reduced by GABA stimulation, consistent with our previous observations[18].

We then expressed SNAP-GABA$_{B1}$ in primary hippocampal neurons at the low levels required for single-particle tracking (0.33 ± 0.08 particles/μm$^2$)—which is far below the relatively high density of GABA$_B$ receptors measured in neurons (~15–100 particles/μm$^2$)[35–37]—together with either a presynaptic (Bassoon-GFP) or postsynaptic (Homer-GFP) marker (Fig. 5a, b). Single-particle tracking followed by a TAMSD analysis revealed accumulation of GABA$_B$ receptors at presynaptic and,

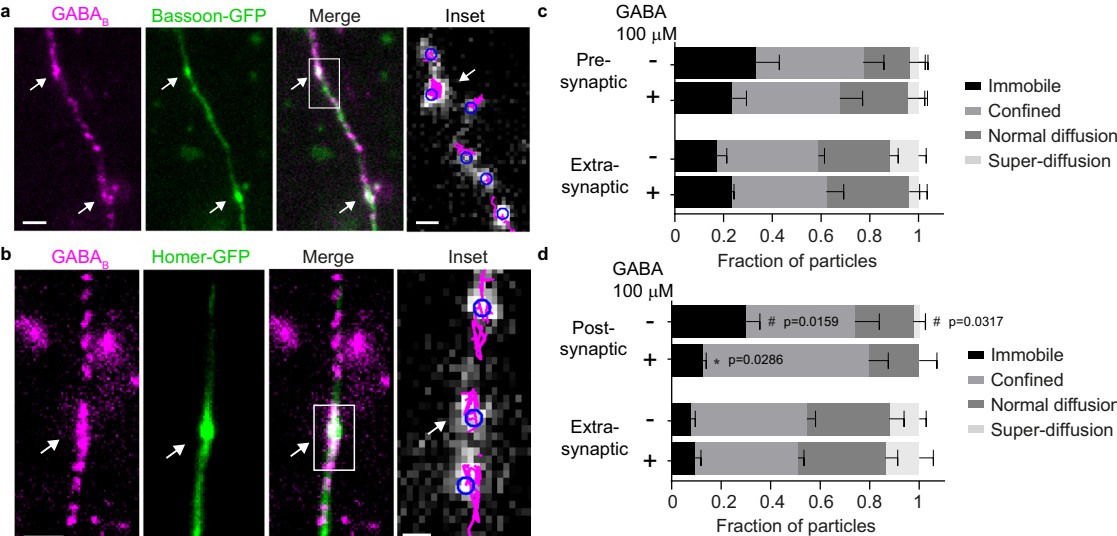

**Fig. 5 | GABA_B receptors are dynamically trapped at synaptic sites in hippocampal neurons. a, b** Representative TIRF images of SNAP-GABA_B1 (magenta) co-transfected with Bassoon-GFP (**a**) or Homer-GFP (**b**) in hippocampal neurons and labeled with SNAP-549 (green). Images are representative of three independent experiments. Arrows indicate colocalization between GABA_B1 and either Bassoon (**a**) or Homer (**b**) at synaptic sites. Insets, results of single-particle tracking applied to the regions delimited by the white boxes. Trajectories are shown as in Fig. 4b. **c, d** Frequency distributions of GABA_B trajectories in hippocampal neurons classified in the four groups identified by the TAMSD analysis in basal and stimulated (GABA 100 μM; 5 min incubation) conditions. Data are obtained from primary hippocampal neurons expressing either Bassoon-GFP (**c**) or Homer-GFP (**d**) to separately analyze the behavior of pre- and postsynaptic GABA_B receptors. Data are mean ± SEM. $n$ = 6 and 4 cells (255 and 82 trajectories) for Bassoon in basal and stimulated condition, respectively, and 5 and 5 cells (74 and 236 trajectories) for Homer in basal and stimulated condition, respectively, examined over three independent experiments. *$p < 0.05$ vs. basal condition and #$p < 0.05$ vs. extra-synaptic compartment by two-tailed Mann-Whitney U test. Scale bars, 2 μm (images in **a, b**) and 500 nm (insets in **a, b**). Source data are provided as a Source Data file.

to a lesser extent, postsynaptic sites, where relevant fractions of the receptors (33% at pre- and 30% at postsynaptic sites) were virtually immobile (Fig. 5c, d and Supplementary Movie 5). GABA stimulation caused a decrease in the immobile fraction (from 30% to 13%) at postsynaptic sites, whereas no statistically significant changes were observed at pre- or extra-synaptic sites (Fig. 5c, d). Pretreatment with a dynamin inhibitor (Dyngo4a) to prevent endocytosis did not change the fraction of immobile GABA_B receptors at postsynaptic sites after GABA stimulation (Supplementary Fig. 5c), indicating that the observed decrease was not due to removal of immobile GABA_B receptors by endocytosis. These results suggest that GABA_B receptors undergo dynamic trapping at both pre- and postsynaptic sites, leading to their local accumulation, which, at least at postsynaptic sites, can be partially released by GABA stimulation.

## FLNA-GABA_B interactions prolong G protein signaling

We then sought to investigate whether, besides controlling the localization of GABA_B on the plasma membrane, GABA_B-FLNA interactions might also impact downstream signaling more directly. For this purpose, we followed G protein signaling in real-time by monitoring FRET between fluorescently labeled Gα_i and Gγ_2 subunits, which dissociate upon G_i protein activation[38,39] (Fig. 6a). Measurements were performed in HEK293A cells, which support the higher plasma membrane expression of membrane proteins required for FRET, co-transfected with GABA_B1, GABA_B2 and either DsRed-FLNA17-18 or DsRed-FLNA19-20. Transient GABA stimulation via a fast superfusion system led to a rapid and reversible decrease of FRET, indicative of G_i protein activation (Fig. 6b). No statistically significant differences were observed in either the amplitude or the kinetics of G_i protein activation between cells expressing FLNA17-18 and FLNA19-20 (Supplementary Fig. 6a–c). In contrast, FLNA19-20 expression was associated with slower deactivation kinetics, indicative of prolonged G_i protein signaling after transient GABA stimulation (Fig. 6c, d). In principle, this effect could be due to either the displacement of GABA_B from actin fibers or the binding of FLNA19-20 to the IL1 of GABA_B1, which might have an

allosteric effect on GABA_B signaling. To further investigate the involved mechanism, we took advantage of M2 cells, a widely used melanoma cell model that lacks endogenous FLNA expression, and A7 cells, a control M2 clone in which FLNA was reintroduced by stable transfection[40]. Similar to what observed in CHO cells, GABA_B showed the characteristic striated pattern and colocalization with F-actin in A7 cells, which were absent in M2 cells (Fig. 6e), further demonstrating that FLNA mediates GABA_B anchoring to the actin cytoskeleton. We then compared the kinetics of G_i protein activation and deactivation in response to transient GABA stimulation in M2 and A7 cells as previously done in HEK293A cells. No statistically significant differences were observed in the amplitude or kinetics of G_i protein activation between M2 and A7 cells (Supplementary Fig. 6d–f). In contrast, G_i protein deactivation was significantly faster in M2 cells that lack FLNA than in control A7 cells (Fig. 6f, g). Altogether, these results indicate that GABA_B-FLNA interactions modify GABA_B signaling by prolonging G_i activation after transient GABA stimulation, likely via an allosteric effect of FLNA binding to the IL1 of GABA_B1 on G_i protein signaling (Fig. 6h).

## Discussion

Our study identifies and characterizes a previously unknown interaction between GABA_B, a crucial mediator of the main inhibitory neurotransmitter in the central nervous system, and FLNA, a key actin cross-linker and scaffolding protein responsible for anchoring membrane proteins to the actin cytoskeleton[41]. This newly uncovered GABA_B-FLNA interaction influences the location and mobility of GABA_B receptors on the plasma membrane in an agonist-dependent manner. Moreover, it prolongs G_i protein signaling after transient GABA stimulation, a condition mimicking intermittent neurotransmitter release, which could provide an additional mechanism to fine-tune GABA_B signaling.

The precise organization of neurotransmitter receptors and other proteins at synaptic sites is essential for rapid and efficient synaptic transmission in the brain[42-45]. These phenomena have been mainly

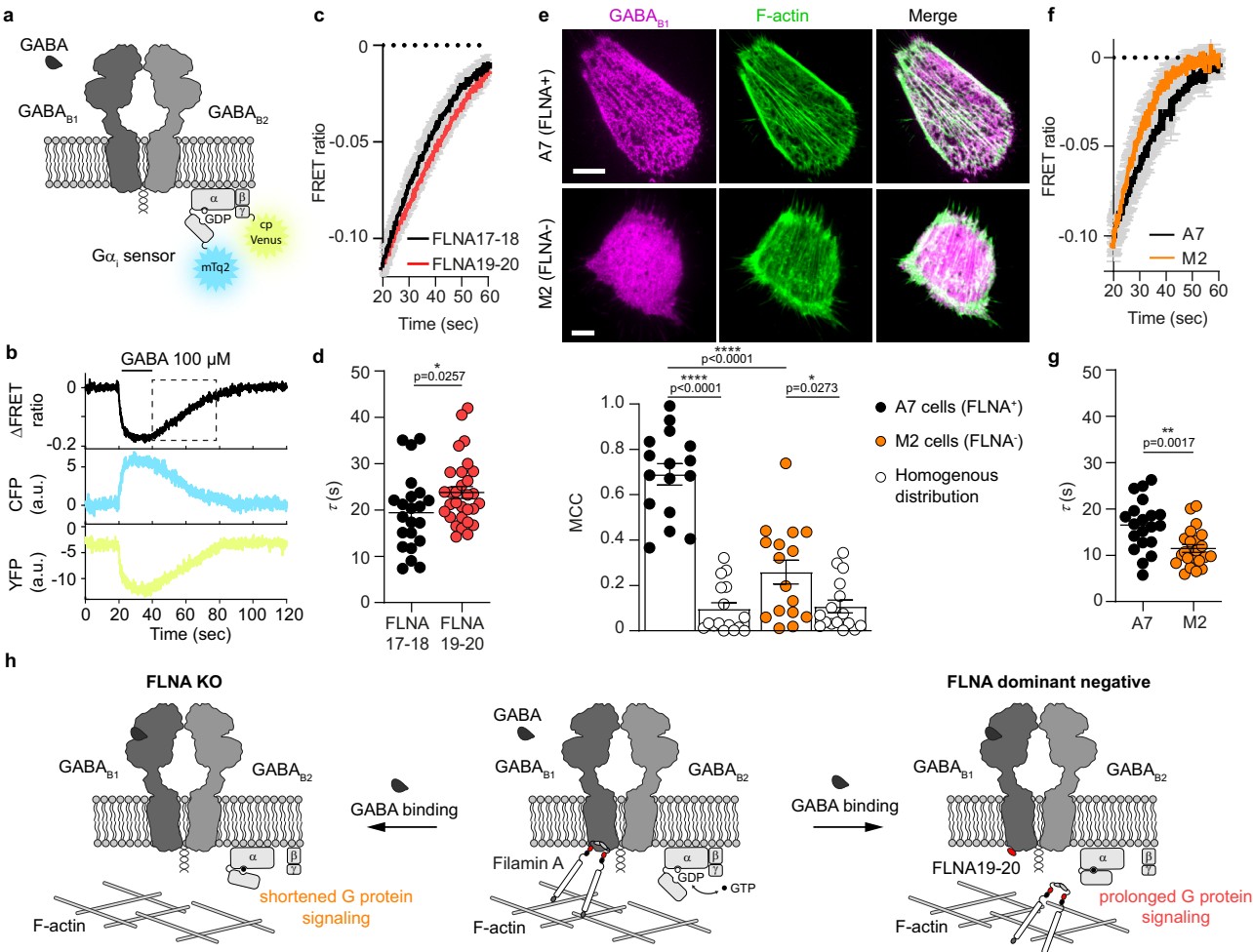

**Fig. 6 | Impact of GABA_B-FLNA interactions on G_i protein signaling. a** Scheme of the FRET sensor[34] used to monitor G_i protein activation in real-time. This consists of a single plasmid encoding G_αi fused to the fluorescent donor mTurquoise2 (mTq2), G_β1, and G_γ2 fused to the fluorescent acceptor cp173Venus (cpVenus). A decrease in FRET is indicative of G_i protein activation. **b** Representative FRET measurement of G_i protein activation and deactivation in response to transient GABA stimulation (100 μM). **c** Comparison of the kinetics of G_i protein deactivation from experiments as in **b** in the presence of FLNA17-18 or FLNA19-20. Data are mean ± SEM. $n$ = 22 and 30 cells (FLNA17-18 and FLNA19-20, respectively) examined over three independent experiments. **d** Corresponding estimated time constants ($\tau$) for G_i protein deactivation. Data are mean ± SEM. $n$ = 22 and 30 cells (FLNA17-18 and FLNA19-20, respectively) examined over three independent experiments. **e** Top, representative images of GABA_B1 (magenta) over-expressed in M2 cells (FLNA⁻) or A7 cells (FLNA⁺) together with Lifeact-GFP (green). Scale bars, 5 μm. Bottom, corresponding MCC

analyses. Data are mean ± SEM. $n$ = 16 and 16 cells (M2 and A7 cells, respectively) examined over two independent experiments. **f** Comparison of the kinetics of G_i protein deactivation in M2 and A7 cells after a transient stimulation with GABA as in **b**. Data are mean ± SEM. $n$ = 23 and 20 cells (M2 and A7 cells, respectively) examined over two independent experiments. **g** Corresponding estimated time constants ($\tau$) for G_i protein deactivation. Data are mean ± SEM. $n$ = 23 and 20 cells (M2 and A7 cells, respectively) examined over two independent experiments. **h** Schematic summary of the results. The interactions of GABA_B1 with FLNA tethers GABA_B receptors to the actin cytoskeleton, controlling their localization on the plasma membrane. In addition, interaction of either full length FLNA or FLNA19-20 with GABA_B1 leads to prolonged G_i signaling after transient GABA stimulation, presumably via an allosteric mechanism. This effect is lost in M2 cells lacking FLNA. *$p$ < 0.05, **$p$ < 0.01, ****$p$ < 0.0001 by two-tailed unpaired Welch's t-test. Source data are provided as a Source Data file.

investigated for ionotropic receptors[46–48]. For instance, studies on α-amino-3-hydroxy-5-methyl-4-isoxazole propionate (AMPA) and N-methyl-D-aspartate (NMDA) glutamate receptors have shown that the actin cytoskeleton is involved in controlling the localization, clustering and/or function of these ionotropic receptors at synapses[49–52]. However, very limited information is available for metabotropic receptors[53–55]. Altogether, our single-particle tracking and expansion microscopy experiments in hippocampal neurons are consistent with a partial accumulation of GABA_B receptors at synaptic sites, where they are prevalently immobile and found in close proximity to FLNA, supporting a role of FLNA in controlling their synaptic location.

Our results indicate that the interaction between FLNA and GABA_B is mediated by the IL1 of the GABA_B1 subunit. Using a synthetic peptide library, we could further restrict the binding motif on GABA_B1 to a short

linear sequence comprising Tyr_614, Arg_619, Tyr_620, and Ile_621, which is distinct from a previously suggested consensus FLNA binding motif[56]. Intriguingly, the newly identified FLNA binding site on GABA_B1 is also unique in that it is located in the IL1. While FLNA interacts with several class A GPCRs through one of their three intracellular loops or C-terminal tail, only a few class C GPCRs have been shown to be interacting with FLNA and exclusively through their C-terminal region[28,56–58]. In addition, we identified a second potential FLNA interaction mediated by the known PDZ binding motif at the C-terminus of the GABA_B2 subunit, which has been shown to interact with several intracellular proteins[20,31,59]. However, our results in intact cells indicate that the PDZ binding motif of the GABA_B2 subunit is neither sufficient nor required to mediate GABA_B anchoring to the actin cytoskeleton, further supporting a critical role of the IL1 of GABA_B1 in mediating the interaction with FLNA.

FLNA contains multiple Ig repeats, several of which have been shown to interact with membrane proteins[60]. Previous studies have already demonstrated that FLNA can directly interact with other GPCRs such as dopamine (D2/D3), angiotensin II (AT1), MAS, and somatostatin (SST2) receptors[26,28,29,56,61]. In the case of the SST2 receptor, a series of previous experiments, including in vitro surface plasmon resonance measurements with repeats 17 to 24, mapped the direct, high-affinity binding site to FLNA19-20[26,28]. Our results indicate that FLNA19-20 can also strongly interact with GABA$_{B1}$ via its IL1, suggesting that the same domain may participate in GABA$_B$-FLNA interactions. However, we cannot rule out that other FLNA repeats might also contribute to GABA$_B$ binding.

Based on our results, we propose that FLNA-GABA$_B$ interactions participate in precisely positioning GABA$_B$ receptors at synaptic nanodomains, thus ensuring that they signal at the correct location during synaptic transmission and, therefore, have a positive, indirect impact on GABA$_B$ signaling. The relevance of this high degree of spatial organization is further supported by our observation that GABA$_B$ confinement is decreased by GABA stimulation both in a simple cell system and in hippocampal neurons, which could contribute to signal desensitization or have more complex effects on GABA$_B$ signaling. In addition, we have uncovered a potential second mechanism, whereby GABA$_B$-FLNA interactions appear to more directly modulate G protein signaling downstream of GABA$_B$ receptors. These two mechanisms are not mutually exclusive but likely cooperate to ensure a precise spatiotemporal control of GABA$_B$ signaling. Further studies will be required to investigate the consequences of this complex spatiotemporal organization on synaptic transmission as well as the role of other factors that, besides FLNA, influence the location and diffusion of GABA$_B$ on the plasma membrane.

The view that FLNA binding might modulate GABA$_B$ signaling via both direct and indirect mechanisms is also consistent with previous findings that GABA$_B$ preferentially form hetero-tetramers and higher order oligomers along actin fibers[18], that GABA$_B$ oligomerization modulates G protein signaling[17], and that GABA$_B$ activation is associated with a rearrangement of the GABA$_{B1}$-GABA$_{B2}$ interface within GABA$_B$ oligomers[17,62]. Since the effect of FLNA on GABA$_B$ signaling appears mediated by its binding to the IL1 of GABA$_{B1}$, our findings could also pave the way to the development of allosteric modulators targeting the IL1 of GABA$_{B1}$ and capable of fine-tuning both GABA$_B$ plasma membrane localization and downstream signaling.

## Methods

### Ethical regulations
All animal work was done according to the regulations of the relevant authority, the government of Lower Franconia, Bavaria, and in accordance with the European guidelines for the care and use of laboratory animals, and the guidelines issued by the University of Bordeaux animal experimental committee.

### Animals
Wild-type adult FVB and C57Bl/6J mice were used for the isolation of primary hippocampal neurons.

### Chemical and reagents
Cell culture reagents, Lipofectamine 2000, and TetraSpeck fluorescent beads were from Thermo Fisher Scientific. The Effectene transfection reagent was from Qiagen. The fluorescent benzylguanine derivative SNAP-Surface 549 (SNAP-549), SNAP-Surface 647 (SNAP-647), and the benzylcytosine derivative CLIP-Cell tetramethylrhodamine-Star (CLIP-TMR) were from New England Biolabs. The GABA$_{B1a/b}$ rabbit polyclonal antibody (B17) was kindly provided by Ryuichi Shigemoto (Institute of Science and Technology, Austria). The Filamin A monoclonal antibody (M01), clone 4E10-1B2 was purchased from Abnova (H00002316-M01), the Bassoon mouse monoclonal antibody was from Enzo Life Sciences (NY, USA) (SAP7F407), the Bassoon rabbit polyclonal antibody was from Synaptic Systems (141003), goat Alexa Fluor-532 anti-mouse (A-11002) and goat Alexa Fluor-647 anti-rabbit (A-21245) were from Thermo Fisher Scientific, goat anti-rabbit IgG was from Sigma Aldrich (AP132), goat anti-rabbit CF568 was from Biotium (20801), the GAPDH mouse monoclonal antibody was from Invitrogen (AM4300), the DsRed mouse monoclonal antibody was from Santa Cruz (sc-390909) and the mouse IgG kappa binding protein coupled to HRP was from Santa Cruz (sc-516102). Dyngo4a was from Abcam (ab120689). All other chemical reagents were from Sigma Aldrich, unless stated otherwise. For the automated solid-phase peptide synthesis and microarray production, amino acids and reagents were purchased from either Iris Biotech or Carl Roth and all solvents were purchased from commercial sources and used without further purification.

### Molecular biology
Plasmids coding for wild-type mouse GABA$_{B1a}$ and GABA$_{B2}$ subunits as well as mouse N-terminal SNAP-tagged GABA$_{B1a}$ (SNAP-GABA$_{B1}$) were kindly provided by Jean-Philippe Pin (Institut de Génomique Fonctionnelle, Université de Montpellier, Montpellier, France) and were previously shown to be functional[63]. The GABA$_{B1}$ ER retention deletion mutant was generated by truncating the C-tail from leucine 905 (just before the ER retention motif RSRR). Plasmids encoding full-length FLNA fused to eGFP (eGFP-FLNA) and FLNA repeats 19-20 or 17-18 (FLNA19-20 and FLNA17-18) fused to DsRed were previously described[29]. Plasmids coding for SNAP- and CLIP-tagged FLNA (SNAP-FLNA and CLIP-FLNA) were generated by replacing eGFP in the first hinge region of the construct coding for eGFP-FLNA[26]. The plasmid encoding the G$_{\alpha i}$ sensor was kindly provided by Joachim Goedhart (University of Amsterdam, Amsterdam, the Netherlands)[38]. The G$_{\alpha i}$ sensor is based on a single plasmid coding for the G$_{\alpha i}$ protein with the fluorescent donor mTurquoise2 fused at its N-terminus, G$_{\beta 1}$, and G$_{\gamma 2}$ with the fluorescent acceptor cp173Venus fused at its C-terminus. The construct allows controlled expression of the different G protein subunits thanks to a viral 2A peptide and an internal ribosome entry site (IRES). A construct coding for Bassoon-GFP was kindly provided by Eckart D. Gundelfinger (Leibniz Institute for Neurobiology, Magdeburg, Germany)[64]. A plasmid coding for Homer-GFP was kindly provided by Shigeo Okabe (Department of Cellular Neurobiology, University of Tokyo, Tokyo, Japan).

### Cell culture and transfection
CHO-K1 cells (Leibnitz-Institut DSMZ) were cultured in phenol red-free Dulbecco's modified Eagle's medium (DMEM)/F-12 supplemented with 10% (v/v) fetal bovine serum (FBS), 100 U/ml penicillin and 0.1 mg/ml streptomycin. For single-molecule experiments, CHO-K1 cells were seeded on clean 24-mm round glass coverslips in 6-well culture plates at a density of $1.8 \times 10^5$ cells per well. Clean coverslips were prepared as described in Sungkaworn et al.[65]. DNA transfection was performed 24 h after seeding using Lipofectamine 2000 and following the manufacturer's protocol.

Human embryonic kidney 293 A (HEK293A) cells (ATCC) were cultured in phenol-red free DMEM medium supplemented with 10% (v/v) FBS, 100 U/ml penicillin and 0.1 mg/ml streptomycin. For FRET experiments, HEK293 cells were seeded on 24-mm round coverslips coated with poly-L-Lysine (10% v/v) in 6-well culture plates at a density of $1.8 \times 10^5$ cells per well. DNA transfection was performed 24 h after cell seeding using Effectene and following the manufacturer's protocol. FRET experiments were performed 48 h after transfection.

The human melanoma cell line M2 (lacking expression of FLNA) and its isogenic cell line A7 (stably expressing full-length FLNA) were a kind gift of Prof. Nakamura (Brigham and Women's Hospital, Boston, MA) to GM. They were cultured in α-minimal essential medium (MEM) supplemented with 8% (v/v) newborn calf serum, 2% (v/v) FBS, 100 U/ml

penicillin and 0.1 mg/ml streptomycin. A7 cells were cultured in the presence of 500 µg/ml G418.

All cells were cultured for a maximum of 20 passages and maintained in a humidified atmosphere at 37 °C and 5% $CO_2$.

None of the used cell lines is listed in the International Cell Line Authentication Committee registry of Misidentified Cell Lines (v.11). M2 and A7 cells were partially authenticated by morphology and immunostaining with a FLNA antibody. Commercial cell lines were not further authenticated.

FLNA knockdown was achieved using custom siRNAs (Horizon) (siRNA #1, sense: 5' CAACUAUGCCAGCCAGAAUUU 3', antisense: 5' AUUCUGGCUGGCAUAGUUGUU 3'; siRNA #2, sense: 5' CCAGAAUCUC CCAUAGCAAUU 3', antisense: 5' UUGCUAUGGGAGAUUCUGGUU 3') transfected with the Dharmafect transfection reagent (Horizon) according to the manufacturer's instructions. A scrambled, no-targeting control siRNA pool was used as control. 72 h after siRNA transfection, the cells were further transfected with SNAP-GABA$_{B1}$ and GABA$_{B2}$ constructs and imaged 48 hours later.

### Primary hippocampal neuron culture

Primary hippocampal cultures were prepared from hippocampi of mouse embryos at embryonic day 18 as previously described[66], with minor modifications. Briefly, pregnant mice were sacrificed by cervical dislocation. Hippocampi (male and female) were dissected in cold Hanks' Balanced Salt Solution supplemented with 7 mM HEPES, 10,000 IU/ml penicillin and 10 mg/ml streptomycin. Hippocampi were dissociated by enzymatic digestion with 0.05 % (v/v) trypsin for 5 min at 37 °C. Neurons were then triturated mechanically by passing through a flame-polished glass Pasteur pipette. Hippocampal neurons were plated on clean 24-mm round glass coverslips precoated with 1 mg/ml poly-D-lysine overnight at 37 °C and cultured in phenol-red-free Neurobasal medium supplemented with 2% (v/v) B27, 2 mM GlutaMAX, 100 U/ml penicillin and 0.1 mg/ml streptomycin. Neurons were cultured at a density of 50,000 cells per well in 12-well plates at 37 °C, 5% $CO_2$, and 95% humidity. One-third of the medium was replaced with fresh medium once a week. For single-molecule TIRF experiments, monolayer cultures (70-80% confluence) were transfected by magnetofection (NeuroMag, OZ Biosciences) at 7-9 days in vitro (DIV) according to the manufacturer's instructions.

### Live-cell SNAP/CLIP labeling

Cells were labeled with a combination of membrane impermeable SNAP substrate (SNAP-549 or SNAP-647) to label the receptor and a cell-permeable CLIP substrate (CLIP-TMR) to label intracellular FLNA. Cells were incubated with 1 µM of each substrate in culture medium without antibiotics for 20 minutes at 37 °C. They were then washed three times with culture medium with 5-min incubation at 37 °C after each wash and immediately imaged.

### TIRF imaging

TIRF imaging was performed on a custom set up based on an Eclipse Ti (Nikon, Tokyo, Japan) equipped with 405 nm, 488 nm, 561 nm, and 640 nm diode lasers (Coherent, Santa Clara, CA), a quadruple band excitation filter, a 100X oil-immersion objective (CFI Apo TIRF 100X, 1.49 NA, Nikon), two beam splitters, four separate EMCCD cameras (iXon DU897, Andor) and a temperature control unit. The objective was maintained at 20 °C using a water-cooled inset and an objective ring connected to a thermostat-controlled water bath. Coverslips were mounted in a microscopy chamber filled with phenol red-free culture medium. Cells were initially identified using bright-field illumination. Then, fine focusing in TIRF-mode was performed using low laser power (3%) to minimize photobleaching. Various laser combinations were used depending on the employed fluorophores, i.e., 488 nm for eGFP, 561 nm for DsRed, SNAP-549 and CLIP-TMR, and 640 nm for SNAP-647. For two-color experiments, images were acquired simultaneously on

the corresponding EMCCD cameras. Image sequences (400 frames) were taken in crop and frame-transfer mode, resulting in an acquisition rate of 35 frames per second.

### Single particle tracking

Single-particle detection and tracking were performed using the u-track software[67] in MATLAB as previously described[18]. The images obtained in different channels were registered against each other using a linear piecewise transformation in MATLAB based on reference images of multicolor fluorescent beads (TetraSpeck; 100 nm size)[25]. The inter-channel localization precision after coordinate registration by linear piecewise transformation was ~20 nm.

### TAMSD analysis

To analyze the motion of receptors and FLNA we computed the time-averaged mean square displacement (TAMSD) of individual trajectories derived from TIRF image sequences as previously described[25,26,33,68]. In order to calculate diffusion coefficients (D), the MSD data were fitted with the following equation:

$$MSD(t) = 4Dt^{\alpha} + 4\sigma_l^2 \tag{1}$$

where t indicates time, $\alpha$ is the anomalous diffusion exponent and $\sigma_l$ is the standard deviation of the Gaussian localization error, which was estimated to be 23 nm. The parameter $\sigma_l$ was estimated experimentally in a previous study[69]. The TAMSD plots shown in Supplementary Fig. 5 were obtained after $\sigma_l$ subtraction. Only trajectories lasting at least 120 frames were analyzed. Since this analysis revealed heterogeneity among particles, trajectories were then categorized according to the diffusion parameters D and $\alpha$. Particles with $D < 0.01 \,\mu m^2 \, s^{-1}$ were considered to be immobile. Normal diffusion was attributed to particles that had $D \geq 0.01 \,\mu m^2 \, s^{-1}$ and $0.75 \leq \alpha \leq 1.25$. Sub-diffusion and super-diffusion were assigned to particles with $D \geq 0.01 \,\mu m^2 \, s^{-1}$ and $\alpha < 0.75$ or $\alpha > 1.25$, respectively.

### Single-molecule colocalization index

We calculated a single molecule colocalization index based on a modification of the method developed in Ibach et al.[70], as previously described[25]. The method is based on a modification of Manders colocalization coefficients (MCCs)[24]. Briefly, we generated a binary mask corresponding to the fibers in the actin channel, and we calculated the number of GABA$_B$ localizations in which the mask in the actin channel is equal to 1. We thus generated a colocalization index, for which values can range from −1 in case of perfect anticorrelation to +1 in case of perfect correlation/colocalization, whereas a value of 0 indicates no colocalization.

### Trapping analysis

To assess the motility of GABA$_B$ and FLNA during their interactions, we applied to each trajectory an algorithm that divides the trajectory into shorter segments corresponding to phases of free diffusion and confinement[34]. This algorithm computes a matrix $D_{ij}$ containing the distance between each pair of positions $\{x_i, y_i\}$ and $\{x_j, y_j\}$ within the trajectory. The matrix $D_{ij}$ is then rescaled by a test length scale accounting for the typical size of expected confinement domains. A binary matrix is constructed where entries are 1 if $\exp(-D_{ij}^2) < 0.36$ or 0 otherwise. In this binary matrix, square blocks of values 1 along the matrix diagonal correspond to trapped portions of the trajectory. To remove false positives, the size of the obtained square blocks is compared to what could be detected by chance for a purely free Brownian motion and excluded if the probability for a Brownian motion to produce blocks of such size is larger than $p = 0.01$. This information was used to classify the GABA$_B$/FLNA colocalizations into 4 groups corresponding to the identified diffusivity state of GABA$_B$ and FLNA during each colocalization.

## Immunofluorescence staining

Hippocampal neurons at 14 DIV were washed twice with phosphate buffered-saline (PBS) for 5 min each and fixed with 4% paraformaldehyde for 15 min at room temperature (RT). Neurons were then washed twice with PBS for 5 min each. Subsequently, the neurons were permeabilized with 0.2 % Triton X-100 in PBS for 5 min at RT and blocked with 5 % goat serum in PBS for 1 h at RT. Samples were then incubated with the appropriate concentrations of primary antibodies in blocking solution overnight at 4 °C. Primary antibodies were used at the following dilutions: rabbit anti-GABA$_{B1a/b}$ (1:400; stock solution concentration = 0.5 mg/ml), mouse anti-Filamin A (1:400; stock concentration = 0.5 mg/ml), monoclonal mouse anti-Bassoon (1:400) and polyclonal rabbit anti-Bassoon (1:400; stock solution = 1 mg/ml). On the next day, neurons were washed with PBS and incubated with the Alexa Fluor-conjugated secondary antibodies diluted 1:200 in blocking solution for 2 h at RT in the dark. F-actin was labeled by incubating the neurons with Alexa Fluor 532 phalloidin for 2 h at RT, followed by two washes with PBS.

## Expansion microscopy (ExM) of immunolabeled primary neurons

Atto643 (NHS-Atto643, Atto-Tec) was conjugated to goat anti-rabbit IgG (Sigma) in 100 mM NaHCO$_3$ using 5-fold molar excess of dye for 2 h at RT, purified using Zeba Spin desalting columns with 40 MWCO (Thermo Fisher Scientific) and stored in PBS containing 0.2 % sodium azide. Goat anti-rabbit CF568 (Biotium, 10 μg/ml) and goat anti-mouse Atto643 (custom labeled, 10 μg/ml) were applied to samples in blocking buffer for 2 h at RT. Following washing in PBS (3x, 10 min each), immunostained samples were reacted with crosslinking reagent AcX (Thermo Fisher Scientific) dissolved at a concentration of 1 μg/ml in PBS overnight at RT. After washing away excess AcX (3x, 10 min each, PBS) samples were gelled as previously described[61,62]. Briefly, coverslips were flipped on 120 μl proExM monomer solution (8.55% sodium acrylate, 2.5% acrylamide, 0.15% bis-acrylamide, 0.2% ammonium persulfate, 0.2% tetramethylethylenediamine) pipetted on parafilm and gelled in a humidified atmosphere at 37 °C for 2 h. Gels were cut to a SIM card shape for the identification of gel orientation and subjected to digestion with 8 U/ml Proteinase K (Thermo Fisher Scientific) in proExM digestion buffer (50 mM Tris pH 8.0, 1 mM EDTA, 0.5% Triton X-100, 0.8 M guanidine HCl) in a 6-well cell culture chamber (3 ml volume) overnight at RT. Digested gels were expanded to the final size by repeated incubations (5-6 times, 15 min each) in double distilled water. Samples were transferred to imaging chambers coated with Poly-D lysine (Merck, high precision glass bottom, 1.5#). Imaging was performed on a LSM700 laser scanning confocal microscope (Zeiss) using a 63X 1.2 NA water objective, equipped with 561 nm and 640 nm DPSS laser lines. Z-Stacks were corrected for chromatic aberration applying elastic transformations based on images of Tetraspeck beads acquired under the same imaging conditions. Image brightness and contrast were linearly adjusted. Confocal images were acquired with ZEN 12.0.1.362 (Zeiss).

## Automated Solid-Phase Peptide Synthesis and Microarray Production

μSPOT peptide arrays[71] (CelluSpots, Intavis AG, Cologne, Germany) containing full length intracellular loops of GABA$_{B1}$ and GABA$_{B2}$ and 15mer-peptides with single amino acid shift corresponding to the C-termini of the receptors, were synthesized using a MultiPep RSi robot (Intavis AG). In-house produced, acid labile, amino functionalized, cellulose membrane discs containing 9-fluorenylmethyloxycarbonyl-β-alanine (Fmoc-β-Ala) linkers (average loading: 130 nmol/disc) were used as a solid support. Synthesis was initiated by Fmoc deprotection using 20% piperidine in dimethylformamide (DMF) followed by washing with DMF and ethanol (EtOH). Peptide chain elongated with a coupling solution (3 eq. to the loading capacity) consisting of preactivated amino acids (aa, 0.5 M), ethyl 2-cyano-2-(hydroxyimino)acetate (Oxyma, 1 M) and N,N'-diisopropylcarbodiimide (DIC, 1 M) in DMF (1:1:1, aa:Oxyma:DIC). Couplings were carried out for 3 × 30 min, followed by capping (4% acetic anhydride in DMF) and washes with DMF and EtOH. Synthesis was finalized by deprotection with 20% piperidine in DMF, and subsequent washes with DMF and EtOH. Dried discs were transferred to 96 deep-well blocks and treated, while shaking, with sidechain deprotection solution, consisting of 90% trifluoracetic acid (TFA), 2% dichloromethane (DCM), 5% H$_2$O and 3% triisopropylsilane (TIPS) (150 μL/well) for 1.5 h at RT. Afterwards, the deprotection solution was removed, the discs were solubilized overnight at RT, while shaking, using a solvation mixture containing 88.5% TFA, 4% trifluoromethanesulfonic acid (TFMSA), 4.5% H$_2$O and 3% TIPS (250 μL/well). The resulting peptide-cellulose conjugates (PCCs) were precipitated with ice-cold ether and spun down at 2000 × g for 10 min at 4 °C, followed by two additional washes of the formed pellet with ice-cold ether. The resulting pellets were dissolved in dimethyl sulfoxide (250 μL/well) to give final stocks. PCC solutions were mixed 2:1 with saline-sodium citrate (SSC) buffer (150 mM NaCl, 15 mM trisodium citrate, pH 7.0) and transferred to a 384-well plate. For transfer of the PCC solutions to white coated CelluSpot blank slides (76 × 26 mm, Intavis AG), a SlideSpotter (Intavis AG) was used. After completion of the printing procedure, slides were left to dry overnight. The resulting on-chip peptide microarray was then used for the biochemical assay with FLNA fragments.

## Peptide microarray binding assay

Cell lysates from HEK293A cells expressing DsRed-tagged FLNA fragments were used to identify the GABA$_B$ domains responsible for FLNA interaction. Briefly, cells were cultured in 10-cm Petri dishes and transfected with the plasmids expressing DsRed-FLNA17-18 or DsRed-FLNA19-20 48 h prior to lysate preparation. The cells were then lysed in 20 mM HEPES, 200 mM NaCl, 0.5 mM EDTA, 10% glycerol, 0.2 % NP-40, pH 7.9, supplemented with protease and phosphatase inhibitors and incubated for 30 min at 4 °C on ice. After thorough mixing, the lysates were centrifuged at 18,000 x g at 4 °C for 10 min. Supernatants were kept at −80 °C until further use.

The microarray chips were blocked at RT for 1 h with 2% bovine serum albumin (BSA) in PBS. Serial dilutions of the supernatants with 0.1% BSA in PBS (1:3, 1:6, 1:12) were applied to the microarray chips and incubated at RT for 1 h. Afterwards the chips were washed with PBS and imaged on an Azure 400 Imaging System (Azure Biosystems, Dublin, USA) with the Cy3 fluorescent filter set. The obtained images were quantified using FIJI (https://fiji.sc/)[72] and the Microarray Profile plugin (OptiNav Inc.).

## Western blot analyses

For the FLNA knockdown experiments (Supplementary Fig. 2a), the cells were lysed with a buffer containing 150 mM NaCl, 1% Triton X-100, 0.5% sodium deoxycholate, 0.1% Sodium Ddodecyl Sulfate (SDS), 50 mM Tris-HCl, pH 8.0, supplemented with protease inhibitors (Roche). The lysates were centrifuged at 10,000 x g for 15 min at 4 °C. The supernatants were mixed with Laemmli buffer, incubated at 95 °C for 5 min, separated by electrophoresis on a 10% SDS polyacrylamide gel and transferred to a PVDF membrane (Millipore). The membrane was blocked with PBS supplemented with 1% Tween and 5% skim milk powder for 1 h at RT and incubated with a mouse anti-FLNA antibody (1:5000) or anti-GAPDH antibody (1:5000) ON at 4 °C, followed by incubation with an anti-mouse HRP-conjugate secondary antibody. Membranes were imaged on a Bio-Rad ChemiDoc system.

To validate the identity of the FLNA-DsRed fragments used in peptide microarray (Supplementary Fig. 4b), cells expressing the fragments were lysed on ice with 0.1% Triton X-100 in PBS supplemented with protease inhibitors (Roche) and centrifuged at 10,000 x g for 15 min at 4 °C. The supernatants were mixed with Laemmli buffer

**Article**

and incubated at 95 °C for 5 min, separated by electrophoresis on a 10% SDS polyacrylamide gel and transferred to a nitrocellulose membrane (Roche). The membrane was blocked for 2–2.5 h with Tris-buffered saline supplemented with 0.1% Tween 20, 1% milk, 3% BSA and incubated with a DsRed antibody (1:1000) ON at 4 °C and afterwards with mouse IgG kappa binding protein conjugated to HRP for 1 h at RT. Membranes were imaged on an Azure 400 system.

## FRET measurements

FRET measurements were performed on an Axiovert 200 inverted microscope (Zeiss) equipped with a 100X oil-immersion objective (Plan-Neofluar 100X, 1.30 NA), a beam splitter (DCLP505), a Poly-chrome IV monochromator and a dual-emission photometric system (Till Photonics). Transfected HEK293A cells were placed in a micro-scopy chamber filled with imaging buffer (137 mM NaCl, 5.4 mM KCl, 2 mM CaCl$_2$, 1 mM MgCl$_2$, 10 mM HEPES, pH 7.3). Illumination was set to 20 ms out of a total cycle of 100 ms. CFP (480 ± 20 nm) and YFP (535 ± 15 nm) signals were recorded simultaneously upon excitation at 436 ± 10 nm. Agonist stimulation was applied using a pressurized rapid superfusion system (ALA-VM8, ALA Scientific Instruments). Fluorescence signals were detected by photodiodes and digitalized using an analogue-digital converter (Digidata 1440 A, Axon Instruments). All data were recorded on a PC running Clampex 10.3 software (Axon Instruments). FRET was expressed as the ratio between YFP and CFP emission upon CFP excitation. The YFP signal was corrected for direct YFP excitation and bleed-through of CFP emission into the YFP channel as previously described[73]. Resulting individual traces were fitted to a one-component exponential decay function.

## Statistical analyses

Data are reported as mean ± SEM. Statistical analyses were conducted using Prism 6 software (GraphPad Software, La Jolla, CA, USA). P-values were determined by two-tailed unpaired Welch's $t$-tests and two-tailed Mann-Whitney U tests to assess differences between two groups. Differences were considered significant for $p < 0.05$.

## Reporting summary

Further information on research design is available in the Nature Portfolio Reporting Summary linked to this article.

## Data availability

Because of their large size (more than 1 Tb), the raw microscopic images underlying the results of our manuscript are available upon request to the corresponding authors. Source data are provided with this paper.

## Code availability

Custom scripts used in the manuscript are available on the GitHub repository: https://github.com/CalebiroLab/GABAB_FLNA.

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

## Acknowledgements

We thank Bianca Klüpfel and Ulrike Zabel for the technical support. We thank Donatella Treppiedi for advice on melanoma cell culture and transfection. We thank the Interdisciplinary Institute for Neurosciences Cell Biology Facility and in particular E. Verdier and S. Daburon for help with neuronal cell culture. We also acknowledge the help of Anna Brachet and Daniel Choquet and his laboratory for providing infrastructure and technical support for some of the imaging experiments. This work was supported by grants from the Deutsche Forschungsgemeinschaft (Sonderforschungsbereich/Transregio 166–Project C1 to D.C., CA 1014/1-1 to D.C., DFG MA6957/1-1 to H.M.M.) and the IZKF Würzburg (grant B-281 to D.C.). M-L.J. was partially funded by a grant from the Fondation pour la Recherche Médicale (FRM ARF201809006996). S.S. was funded by a grant of the German Excellence Initiative and by the DAAD STIBET program through the Graduate School of Life Sciences, University of Würzburg. D.C. is supported by a Wellcome Trust Senior Research Fellowship (212313/Z/18/Z).

## Author contributions

D.C. and G.M. conceived the study. D.C. and M-L.J. designed the TIRF and FRET experiments. M-L.J., Z.K., T.S., K.S., and C.M. performed the TIRF and FRET experiments, and M-L.J., Y.L., and C.M. analyzed the data. TIRF and confocal experiments on neurons were designed by D.C., S.S., and M-L.J., performed by S.S., M-L.J., and C.M., and analyzed by M-L.J., S.S., and Y.L. Micro-array assays were designed by H.M. and V.K. and performed by V.K. The expansion microscopy experiments were designed by D.C., M.S., M-L.J., S.S., and C.W. C.W. performed the experiments and M-L.J. and C.W. analyzed the data. D.C. supervised the study. D.C. and M-L.J. wrote the manuscript. All authors discussed the results and reviewed the manuscript.

## Competing interests

The authors declare no competing interests.
