## [Peer Review File · Nature Communications]

Filamin A organizes γ -aminobutyric acid type B receptors at the plasma membraneREVIEWER COMMENTS

Reviewer #1 (Remarks to the Author):

This article describes the spatial organization and diffusion of the GABAB receptor, a family C GPCR, regulated by dynamic interactions with filamin A. First, the authors confirmed the previous observation that GABAB receptors partially arranged in rows via interaction with the actin cytoskeleton in CHO cells by using TIRF microscopy. Then, they identified filamin A (FLNA) as a scaffold protein tethering the GABAB receptors to actin fibers. They also observed the colocalization between GABAB receptors and FLNA in the primary mouse hippocampal neurons by using expansion microscopy. From the results of TIRF imaging of GABAB receptor mutants and peptide microarray binding assays, the authors concluded that four amino acid residues in the intracellular loop 1 (IL1) in the GB1 subunit of GABAB receptor are responsible for the interaction with FLNA. Single-molecule analysis in CHO cells suggested that the mobility of GABAB receptors changes in the presence of a dominant-negative filamin A fragment (FLNA19-20). The mobilities of GABAB receptors were compared intra- and extra-synapses of the hippocampal neurons. A GABA stimulation-dependent change in the frequency and duration of colocalization between GABAB receptors and FLNA was also found. Finally, the authors assessed the effect of GABAB receptor-FLNA interaction on the kinetics of Gi protein activation by using a FRET sensor in HEK293 cells.

Regarding the identification of the interaction sites between GABAB receptor and FLNA, this manuscript is well designed and provides novel insight. Especially, it is an unexpected finding that the IL1 of the GB1 subunit is responsible for the interaction with a scaffold protein. It is also excellent that the imaging analyses were performed not only in cultured cells but also in neurons where GABAB receptors function physiologically. On the other hand, the data provided are insufficient on how FLNA-mediated changes in the spatial organization of GABAB regulate their activity. Some revisions are required to clarify the relationship between the function and mobility of GABAB as listed below.

Major questions

1) How does FLNA change the mobility of GABAB receptors?

The authors showed that the presence of FLNA19-20 caused a modest decrease in the fraction of immobile receptors (ExtDataFig. 4b) and an increase of the average diffusion coefficient in the normal and super diffusion states (Fig. 4c). However, it is unclear which diffusion state of the GABAB receptors is interacting with FLNA. The authors also performed a two-color single-molecule imaging analysis of GABAB receptor and FLNA (ExtDataFig. 5), however, no data was shown regarding the mobility of GABAB interacting with FLNA. The Movie 3 and ExtDataFig. 5a provides an insight that the immobile FLNA, which seems to be interacting with an actin fiber, transiently captured the freely diffusing GABAB receptor, but this is only an example of a single molecule. Is there a case where a diffusing FLNA and GABAB receptor interact and then become immobile? To clarify how FLNA change the mobility of GABAB receptors, the authors are encouraged to show the distribution of the four diffusion states (or

step-size/diffusion coefficient distribution) of GABAB receptor that interacting with FLNA. If the receptors that interact with FLNA are mostly immobile, it would provide more supportive evidence for the authors' discussion.

2) How does GABA stimulation change the spatial organization of GABAB receptors in hippocampal neurons?

The authors showed that GABA stimulation decreased the frequency and duration of GABAB-FLNA colocalization (ExtDataFig. 5). These data are essential for understanding the functional importance of GABAB-FLNA interaction, and it would be preferable to move them to the main figure. If these changes have physiological meanings, one can expect to see a GABA-dependent decrease in the percentage of immobile GABAB receptors localized at the synapses. In Fig. 4d-g, the authors only compared the mobility of GABAB receptor localized in the intra- and extra-synapses in the unstimulated condition. This does not allow the readers to know the contribution of the GABAB-FLNA interaction to the GABA-induced changes in the spatial organization of GABAB receptors. Since the synaptic localization of the GABAB receptor would be also decreased by endocytosis, the authors are encouraged to compare GABA-dependent changes in GABAB receptor mobility in neurons overexpressing FLNA19-20 or FLNA17-18.

3) To what extent does the interaction with FLNA contribute the change in mobility of GABAB receptors upon GABA stimulation.

The authors described “These results indicate that agonist stimulation largely abolishes GABAB interactions with FLNA, consistent with our previous observation that GABA stimulation increases the lateral mobility of GABAB receptors (l.291-4)”. This might be a reason for the differences in the activation-dependent mobility changes among GPCRs reported in the previous studies. However, the diffusion dynamics of GPCRs would not be solely determined by the interaction with FLNA. The authors and other groups have reported that the dimerization, the interaction with G proteins, arrestins, and clathrins also affect the mobility of various GPCRs (e.g. Dijkman et al. Nat Commun 2020, Möller et al. Nat Chem Biol 2020, Yanagawa et al. Sci Signal 2018, Sungkaworn et al. Nature 2017, Tabor et al. Sci Rep 2016.). To clarify how the decrease in the interaction with FLNA affected the mobility of total GABAB receptors, the authors should provide the data regarding the diffusion state fractions and diffusion coefficients of each state estimated from the same cells shown in ExtDataFig 5. In addition, Möller et al. demonstrated that the diffusion coefficient of opioid receptors changes with time after agonist stimulation. The authors should clarify the time range of GABA stimulation during which they did the imaging.

4) What is the functional role of the interaction of GABAB with actin fibers?

In the final section, the authors evaluated the GABAB-FLNA interaction on Gi protein signaling in HEK293 cells (Fig. 5). As described by the authors, the data suggested that the binding of FLNA to IL1 of GB1

subunit itself allosterically prolonged the deactivation kinetics of GABAB receptors. In this case, what is the functional role of FLNA for the alignment of the GABAB receptors on the actin fibers? Although the GB1-IL1-mediated allosteric regulation of the activity of GB2 is an interesting new finding, it does not seem to be well linked to the spatial organization and mobility of GABAB receptors shown in the former part of the paper. The authors are encouraged to add a discussion regarding this point.

Minor comments

5) In the "Methods" section (TIRF imaging, lines 618-629, p28), the authors described the protocol for four-color TIRF imaging. However, the present manuscript seems to contain only two-color imaging. The authors should describe the number of lasers and cameras used for the measurements.

6) The "Methods" section contains few protocols for the parameter analysis of the imaging data. The authors should describe them. Especially, the definition of diffusion coefficient, colocalization, and parameters of clustering analysis should be clearly described because they are varied between papers.

Reviewer #2 (Remarks to the Author):

This manuscript from Jobin and Calebiro reveals that a class C GPCR, the GABAB receptor, interacts with filamin via its intracellular loop 1 (IL1). This interaction results in a striking striated organisation of the receptor in cells due to tethering to the actin cytoskeleton. Single particle tracking experiments show that GABAB receptor interaction with filamin controls receptor diffusion and that receptor mobility changes following receptor stimulation, which corresponds with a loss of the filamin interaction. Finally, the authors show that the GABAB-filamin interaction prolongs G protein activity following transient stimulation of the GABAB receptor.

This is a beautiful piece of work, with very clear and well-organised figures, and a great narrative. The methodology is clearly described and robust. The authors have used an impressive array of high-resolution approaches to make these observations, and their conclusions clearly show the importance of receptor organisation in efficient cell responses. This is an important study for the GPCR field.

I have a few questions and comments for the authors to address:

1. Start of results section on pg 10 line 197: "When SNAP-tagged GABAB2 was transfected alone there was no striated pattern". The authors conclude from this that the GABAB2 subunit does not control

filamin/actin localisation. However, I can't see that the authors did the reverse experiment to test whether the striated pattern was still present when the GABAB1 subunit was transfected alone.

There is high homology between the intracellular loops of the two GABAB subunits, and the authors found that replacing the loops of GABAB1 with those of GABAB2 did not change the striated organisation. This seems to suggest to me that these regions of GABAB1 and GABAB2 are interchangeable. In contrast, replacement of the intracellular loops of GABAB1 with those of the mGluR2 removed the patterning.

As currently described, it is not clear to me why the authors conclude that the GABAB2 does not localise to filamin/actin and that this interaction is only mediated by GABAB1. Is it possible that both GABA subunits can control localisation?

Please show whether (or not) transfection of GABAB1 alone results in a striated organisation and discuss these conclusions in more detail. Replacement of the GABAB2 IL1 with that of mGluR2 could also answer this question.

2. Please show the complete G protein activity time courses for all conditions (at least as supplementary figures) in addition to the inset panels currently displayed as Fig 5c and f, and summarised as estimated time constants in Fig 5d and g. This will allow for stronger contrast between the effects on G protein activation vs deactivation.

3. Can you please comment in the discussion on the effect of transfection of SNAP-GABA subunits into primary hippocampal neurons. Would you hypothesise that endogenous populations be even more restricted in their distribution/location? Based on your experiments, is there a large difference in protein expression for exogenous vs endogenous GABAB receptors in neurons?

4. Cartoon of conclusions (Fig 5h) would be clearer if there was an arrow pointing from the first to the second panel to indicate a sequence of events. I also think FLNA19-20 in the second panel should be replaced with the Filamin A cartoon from the first panel – to show what happens in a “normal cell”. You can then make it clear in the legend that FLNA19-20 can substitute for Filamin A. Consider even labelling the third panel as 5i, to be clear that this is what happens in the absence of filamin (e.g. M2 cells).

Minor comments:

1. Labelling of MCC graphs – please re-label some of the MCC graphs to make it clearer what conditions the co-incident detection quantification is referring to.

e.g. the legend in Fig 2h “GABAB1 (vs Actin)” is easier to interpret than the current legends of Fig 2e

2. How many independent transfections/SNAP labellings does the number of cells (e.g. Fig 2b legend states n=11) come from? Please also include this information in the figure legend.

3. Can you please rationalise the switch to HEK293 cells from CHO cells for FRET experiments examining G protein signalling in the results section.

4. The specificity of the peptide array experiments is quite striking. I was wondering whether you could add the normalised data for these experiments as a supplementary data table, so the reader can get an indication of what is designated a high, medium or low interaction on the heat map.

Reviewer #3 (Remarks to the Author):

The actin cross-linking and scaffolding protein filamin has been implicated in connecting a range of membrane proteins to the cytoskeleton. Here, Jobin et al., report that they have extended that list to include GABA-B. While the report contains some potentially interesting observations, evidence for the central conclusion – that GABA-B binds full-length filamin – is weak. Beyond co-localization, no data on GABA-filamin protein-protein interactions are provided. The binding site on FLNA is assumed to be FLNA9-20 although other sites were not examined and other than a peptide array FLNA19-20 is not shown to bind GABA. The filamin-GABA interaction is central to the argument being made and needs to be much more rigorously established. The functional relevance of the interaction is also poorly developed and appears to be modest at best. The authors begin focusing on GABA localization but, if I understand correctly, they conclude that rather than being importance for localizing the protein, filamin binding is important in regulating GABA signaling. The authors should draw more on their point mutated GABA to support their conclusions and I suggest that FLNA knockout or knockdown be employed to a greater extent to validate conclusions.

Specific comments:

Building on their prior results, the authors show a remarkably strong co-localization of SNAP-GABA-B1 with F-actin (Life-act) in CHO cells (Fig 1a-c). However, they do not include a negative control for this experiment, only showing a calculated homogenous distribution. The data should be validated using an irrelevant SNAP-tagged membrane protein. The dynamic colocalization analysis (Fig 1d,e) indicates that control proteins (CD86) may not behave like a model homogeneous distribution highlighting the need for experimental controls.

In seeking potential linkers to the cytoskeleton the authors tested FLNA. FLNA colocalizes with F-actin so, as the authors have already shown GABA co-localization with F-actin, it is unsurprising that FLNA and GABA also colocalize. This co-localization in transfected cells or neurons need not implicate FLNA in the link. Again, a negative control SNAP tagged protein is missing.

The authors attempt to further implicate FLNA by expressing a small fragment of FLNA19-20 with the idea that it might act as a dominant negative fragment. This strategy is difficult to justify – FLNA is reported to bind many proteins and while binding sites do seem to be somewhat enriched in the rod 2 region, selecting only 2 of the total 24 repeats from filamin, especially in the absence of clear data that GABA binds that region, is questionable. Likewise, the statement that that FLNA17-18 are not involved in membrane protein interaction is incorrect (e.g. PMID 16293600, but many other partners may engage FLNA17). Why have the authors not reported the impact of loss of FLNA on GABA localization?

In the images shown in Fig 2d, GABA-B1 seems to be totally mislocalized by the over-expressed FLNA19-20, however quantification results in an average MCC of ~ 0.5 . Is the image shown representative? This experiment also requires additional controls, e.g., does FLNA19-20 alter the localization of other actin-binding membrane proteins?

Using chimeric receptors, the authors convincingly show that the intracellular loop 1 is important for co-localization of GABA-B1 with actin. However, the conclusion that the B2 receptor is not involved (extended data) is compromised by its expression in the absence of B1. Does B1 expressed alone target to actin? Does mutagenesis of the B2 loop – in the context of the heterodimer – impact localization?

To show association with filamin the authors turned to peptide arrays and the FLNA19-20 fragment. Specificity of the peptide arrays is based in part on comparison of the FLNA19-20 and FLNA17-18 proteins but no validation that these proteins (expressed as DsRed fusions) were stably expressed at the expected sizes and at comparable levels was provided.

Using the peptide array, point mutations in GABA-B1 that impact binding were detected but these potentially useful mutations were not employed in any functional experiments.

The use of the M2 and A7 cell lines supports a role for filamin but these lines are problematic in many ways. Notably, they were generated a long time ago and have accumulated differences in protein expression over time making, comparison of phenotypes difficult. The use of knockout cells (or knockdown cells) acutely reconstituted with FLNA or FLNA lacking the 19-20 would provide a more robust approach.

Response to reviewers

Reviewer #1 (R1)

(Remarks to the Author):

This article describes the spatial organization and diffusion of the GABAB receptor, a family C GPCR, regulated by dynamic interactions with filamin A. First, the authors confirmed the previous observation that GABAB receptors partially arranged in rows via interaction with the actin cytoskeleton in CHO cells by using TIRF microscopy. Then, they identified filamin A (FLNA) as a scaffold protein tethering the GABAB receptors to actin fibers. They also observed the colocalization between GABAB receptors and FLNA in the primary mouse hippocampal neurons by using expansion microscopy. From the results of TIRF imaging of GABAB receptor mutants and peptide microarray binding assays, the authors concluded that four amino acid residues in the intracellular loop 1 (IL1) in the GB1 subunit of GABAB receptor are responsible for the interaction with FLNA. Single-molecule analysis in CHO cells suggested that the mobility of GABAB receptors changes in the presence of a dominant-negative filamin A fragment (FLNA19-20). The mobilities of GABAB receptors were compared intra- and extra-synapses of the hippocampal neurons. A GABA stimulation-dependent change in the frequency and duration of colocalization between GABAB receptors and FLNA was also found. Finally, the authors assessed the effect of GABAB receptor-FLNA interaction on the kinetics of Gi protein activation by using a FRET sensor in HEK293 cells.

Regarding the identification of the interaction sites between GABAB receptor and FLNA, this manuscript is well designed and provides novel insight. Especially, it is an unexpected finding that the IL1 of the GB1 subunit is responsible for the interaction with a scaffold protein. It is also excellent that the imaging analyses were performed not only in cultured cells but also in neurons where GABAB receptors function physiologically. On the other hand, the data provided are insufficient on how FLNA-mediated changes in the spatial organization of GABAB regulate their activity. Some revisions are required to clarify the relationship between the function and mobility of GABAB as listed below.

We thank Reviewer 1 for stating that our manuscript is overall well designed and provides novel insights into GABA_B/FLNA interactions as well as for the thorough feedback and constructive suggestions, which have helped us further improve our study. Following the Reviewer's suggestions, we have performed a series of additional experiments, including in hippocampal neurons, which further support our conclusions (see below for details).

Regarding the relationship between GABA_B receptor spatial organization and function, the Reviewer raises an important point, which we feel we may not have sufficiently discussed in our initial submission. It is well known from work on other synaptic receptors that synaptic transmission requires a highly organized spatial organization of receptors at synapses, which is typically achieved via interactions with the underlying cytoskeleton. Therefore, it appears highly plausible that the FLNA-GABA_B interactions uncovered in our study participate in correctly positioning GABA_B receptors at synapses, and, thus, (indirectly) impact on GABA_B signalling without necessarily modulating GABA_B receptor function directly. This view is supported by our expansion microscopy results in hippocampal neurons showing that a relevant fraction of GABA_B is in close proximity to FLNA within diffraction-limited nanodomains prevalently located on dendritic shafts and fine neuronal protrusions. Moreover, it is reinforced by the results of our new experiments (see below), which show that the mobility of GABA_B receptors is regulated by GABA also in hippocampal neurons. In addition to this mechanism,

which might be the predominant one given the fundamental role of spatial organization in synaptic transmission, we have uncovered a potential second mechanism, whereby GABA_B-FLNA interactions appear to more directly fine-tune G protein signalling initiated by GABA_B receptors. These two mechanisms are not mutually exclusive but likely cooperate to ensure a precise spatiotemporal control of GABA_B signalling. We have revised the Discussion to explain these important points (see page 22, Lines 457-478).

Major questions

R1.1) *How does FLNA change the mobility of GABAB receptors? The authors showed that the presence of FLNA19-20 caused a modest decrease in the fraction of immobile receptors (ExtDataFig. 4b) and an increase of the average diffusion coefficient in the normal and super diffusion states (Fig. 4c). However, it is unclear which diffusion state of the GABAB receptors is interacting with FLNA. The authors also performed a two-color single-molecule imaging analysis of GABAB receptor and FLNA (ExtDataFig. 5), however, no data was shown regarding the mobility of GABAB interacting with FLNA. The Movie 3 and ExtDataFig. 5a provides an insight that the immobile FLNA, which seems to be interacting with an actin fiber, transiently captured the freely diffusing GABAB receptor, but this is only an example of a single molecule. Is there a case where a diffusing FLNA and GABAB receptor interact and then become immobile? To clarify how FLNA change the mobility of GABAB receptors, the authors are encouraged to show the distribution of the four diffusion states (or step-size/diffusion coefficient distribution) of GABAB receptor that interacting with FLNA. If the receptors that interact with FLNA are mostly immobile, it would provide more supportive evidence for the authors' discussion.*

We thank Reviewer 1 for raising this important point. We agree that it would be very useful to show the different states in which GABA_B and FLNA are interacting. The TAMSD analysis that we had presented in our initial submission is suitable to assess the average diffusive behaviour of whole trajectories. However, individual GABA_B and FLNA molecules typically alternate between different diffusion states and their interactions are only transient, which complicates the analysis. To overcome these limitations, we have recently developed a novel statistical approach (trapping analysis described in the “Methods” section), which allows one to robustly segment single-molecule trajectories into phases of free diffusion and confinement. Motivated by the Reviewer’s suggestion, we have now applied our new approach to divide GABA_B and FLNA trajectories into smaller segments and measure the diffusion states of GABA_B and FLNA when they colocalize. Different populations of GABA_B and FLNA were observed: a co-confined population where GABA_B and FLNA are both trapped (usually on actin filaments) and colocalize; a co-diffusing population where they colocalize while diffusing; a population of freely diffusing GABA_B that transiently co-localize with trapped FLNA and a population of trapped GABA_B that transiently colocalize with freely diffusing FLNA. Importantly, we observed that GABA_B and FLNA are mainly trapped when they colocalize as correctly predicted by the Reviewer, further supporting our conclusions. A smaller proportion of GABA_B (or FLNA) particles are freely diffusing while they encounter a trapped FLNA (or GABA_B) particle. Upon GABA stimulation we observe a decrease in most populations of colocalized GABA_B and FLNA, whereas no effect is observed in the case of the GABA_B-IL1 (mGluR2) mutant used as control. These new results are now presented in Fig. 4g (see below) and in the “Results” section (Page 14, Lines 305-320).

Figure 4g. Diffusivity states of single-molecule colocalizations between FLNA and either GABA_{B1} or $\text{GABA}_{\text{B1-IL1}}$ (mGluR2) under basal and stimulated conditions (GABA; 100 μM ; 5 min incubation). Data are mean \pm SEM of $n=52$ (GABA_{B1} basal), 20 (GABA_{B1} stimulated), 23 ($\text{GABA}_{\text{B1-IL1}}$ (mGluR2) basal) and 11 ($\text{GABA}_{\text{B1-IL1}}$ (mGluR2) stimulated) cells from three independent experiments. * $p < 0.05$, and ** $p < 0.01$ by two-tailed unpaired t-test.

R1.2) How does GABA stimulation change the spatial organization of GABAB receptors in hippocampal neurons?

The authors showed that GABA stimulation decreased the frequency and duration of GABAB-FLNA colocalization (ExtDataFig. 5). These data are essential for understanding the functional importance of GABAB-FLNA interaction, and it would be preferable to move them to the main figure. If these changes have physiological meanings, one can expect to see a GABA-dependent decrease in the percentage of immobile GABAB receptors localized at the synapses.

In Fig. 4d-g, the authors only compared the mobility of GABAB receptor localized in the intra- and extra-synapses in the unstimulated condition. This does not allow the readers to know the contribution of the GABAB-FLNA interaction to the GABA-induced changes in the spatial organization of GABAB receptors.

We thank the Reviewer for these comments and suggestions. We agree that the results previously shown in ED Fig. 5 are important to support the conclusions of our manuscript and we have therefore moved them to main Fig. 4.

In order to further strengthen our conclusions, we have performed a complete new set of single-molecule experiments in hippocampal neurons comparing GABA_{B} mobility with/without GABA stimulation. Importantly, we observe a decrease in the immobile fraction of GABA_{B} receptors after GABA stimulation, which is particularly evident and reaches statistical significance at post-synaptic sites (vs. pre-synaptic ones) and did not occur at extra-synaptic sites. These results further strengthen our conclusions, providing evidence that the observed GABA effect on GABA_{B} mobility occurs also in hippocampal neurons, and is therefore of potential physiological relevance. These new results are now included in the main figures Fig. 4j and 4k (see below) and “Results” section (Page 15, lines 328-330).

Figure 4j, k. Frequency distributions of GABA_B trajectories in hippocampal neurons classified in the four groups identified by the TAMSD analysis in basal and stimulated (GABA 100 μ M; 5 min incubation) condition. Data are obtained from primary hippocampal neurons expressing either Bassoon-GFP (**j**) or Homer-GFP (**k**) to separately analyze the behavior of pre- and postsynaptic GABA_B receptors. Data are mean \pm SEM of n=5 and 5 cells (255 and 82 trajectories) for Bassoon in basal and stimulated condition, respectively, and 6 and 4 cells (74 and 236 trajectories) for Homer in basal and stimulated condition, respectively, from three independent experiments. * $p < 0.05$ vs. basal condition and # $p < 0.05$ compared to the extra-synaptic compartment by Mann-Whitney U test.

Since the synaptic localization of the GABA_B receptor would be also decreased by endocytosis, the authors are encouraged to compare GABA-dependent changes in GABA_B receptor mobility in neurons overexpressing FLNA19-20 or FLNA17-18.

We thank Reviewer 1 for raising this point. Single-molecule experiments in hippocampal neurons are rather challenging. Although we were able to perform co-transfection of up to three plasmids in neurons, the suggested experiment, i.e. overexpressing a fourth plasmid (FLNA17-18 or FLNA19-20) in hippocampal neurons at levels sufficient to cause a robust dominant negative effect but, importantly, without causing toxicity, followed by sensitive single-particle tracking analyses in the very few cells expressing all constructs, is not feasible in our hands. We have therefore resorted to a different approach to address the question raised by the Reviewer and exclude a relevant contribution of endocytosis on the observed changes in GABA_B synaptic localization. To this aim, we have performed new experiments in hippocampal neurons pre-treated with Dyngo4a, a well-established dynamin inhibitor and endocytosis blocker, which has already been employed in hippocampal neurons due to its effectiveness and low toxicity. Importantly, Dyngo4a pre-treatment did not cause any changes to the mobility of GABA_B receptors at pre-synaptic sites after GABA stimulation, ruling out a relevant contribution of endocytosis in the observed GABA-dependent decrease of the immobile GABA_B receptor fraction. We have added these new results in ED Fig. 5c and in the corresponding “Results” section (Page 15, lines 328-336).

Extended Data Figure 5c. Frequency distributions of GABA_{B1} trajectories in hippocampal neurons pre-treated with Dyngo4a (30μM) and stimulated for 5 min with GABA (100 μM), classified in the four diffusivity groups. Data are mean ± SEM of n=4 and 14 cells (236 and 255 trajectories) without or pre-treated with Dyngo4a, respectively, from three independent experiments.

R1.3) To what extent does the interaction with FLNA contribute the change in mobility of GABAB receptors upon GABA stimulation.

The authors described “These results indicate that agonist stimulation largely abolishes GABAB interactions with FLNA, consistent with our previous observation that GABA stimulation increases the lateral mobility of GABAB receptors (1.291-4)”. This might be a reason for the differences in the activation-dependent mobility changes among GPCRs reported in the previous studies. However, the diffusion dynamics of GPCRs would not be solely determined by the interaction with FLNA.

The authors and other groups have reported that the dimerization, the interaction with G proteins, arrestins, and clathrins also affect the mobility of various GPCRs (e.g. Dijkman et al. Nat Commun 2020, Möller et al. Nat Chem Biol 2020, Yanagawa et al. Sci Signal 2018, Sungkaworn et al. Nature 2017, Tabor et al. Sci Rep 2016.). To clarify how the decrease in the interaction with FLNA affected the mobility of total GABAB receptors, the authors should provide the data regarding the diffusion state fractions and diffusion coefficients of each state estimated from the same cells shown in ExtDataFig 5.

We thank Reviewer 1 for highlighting this important point. It is indeed well established that the diffusion dynamics of membrane proteins, including GABA_B and other GPCRs, is affected by multiple factors, including the ones mentioned by the Reviewer. Whereas the newly uncovered interaction with FLNA identified in our study appears to play a key role in tethering GABA_B to the underlying actin cytoskeleton, there are certainly other factors that influence GABA_B diffusion. As detailed in our response to the Reviewer’s previous point (see R1.2), we have now performed a more thorough analysis (shown in Fig. 4g) of the diffusive states of GABA_B and FLNA molecules during their colocalization, which is based on the same data presented in Fig. 4d-f (previously ED Fig. 5). As explained in detail above, this analysis revealed that the majority of colocalizations occur while both molecules are confined/trapped (which typically occurs on actin filaments). Average diffusion coefficients are not very informative in this case, as what really matters is the change in the trapped fraction. Overall, these results further confirm that the interactions with FLNA have a clear impact on the localisation/diffusion of GABA_B and are regulated by GABA. A sentence clarifying that, in addition to FLNA, several other factors likely influence GABA_B diffusion has been added to the Discussion (see Page 22, lines 475-478).

In addition, Möller et al. demonstrated that the diffusion coefficient of opioid receptors changes with time after agonist stimulation. The authors should clarify the time range of GABA stimulation during which they did the imaging.

We thank the Reviewer for requesting this important information, which was inadvertently overlooked in the initial submission. In our experiments, we have used a short-time stimulation with GABA for 5 min, which is sufficient to induce full GABA_B activation, without triggering substantial CCP accumulation and internalization. The latter phenomena are largely responsible for the typical time-dependent changes in GPCR diffusion that we and others have reported in previous studies. As the focus of the present study was on the mechanisms governing GABA_B anchoring to the actin cytoskeleton and not on the (later) events involved in GABA_B internalization, the stimulation time was carefully chosen to avoid these potentially confounding effects. The stimulation time has now been clarified in the main text (Page 14, Line 301 and following).

R1.5) *What is the functional role of the interaction of GABAB with actin fibers? In the final section, the authors evaluated the GABAB-FLNA interaction on Gi protein signaling in HEK293 cells (Fig. 5). As described by the authors, the data suggested that the binding of FLNA to IL1 of GB1 subunit itself allosterically prolonged the deactivation kinetics of GABAB receptors. In this case, what is the functional role of FLNA for the alignment of the GABAB receptors on the actin fibers? Although the GB1-IL1-mediated allosteric regulation of the activity of GB2 is an interesting new finding, it does not seem to be well linked to the spatial organization and mobility of GABAB receptors shown in the former part of the paper. The authors are encouraged to add a discussion regarding this point.*

We thank the Reviewer for raising this important point. As discussed in our response to the Reviewer's general comments (see above), we believe that there are at least two distinct mechanisms through which FLNA can modulate GABA_B function. One is by controlling its precise location at synaptic sites via tethering it to actin fibres, which has been shown for many other synaptic receptors to be crucial for efficient synaptic transmission. Of note, our study identifies and characterized the mechanisms responsible for this important interaction of GABA_B with the actin cytoskeleton. The second mechanism involves an additional "allosteric" effect that we have uncovered whereby the interaction with FLNA appears to more directly modulate G protein signalling downstream of GABA_B. The two mechanisms are distinct, and as far as we know not linked, but likely cooperate to ensure a precise spatiotemporal control of GABA_B signalling. These important points are now clarified in the Discussion (Page 22, lines 466-478).

Minor comments

R1.6) *In the "Methods" section (TIRF imaging, lines 618-629, p28), the authors described the protocol for four-color TIRF imaging. However, the present manuscript seems to contain only two-color imaging. The authors should describe the number of lasers and cameras used for the measurements.*

We thank Reviewer 1 for this request for clarification. The used 4-camera set-up is capable of up to 4 colours simultaneously, but only 2-colour experiments were performed in this study. We have included the suggested modifications in the "Methods" section (Pages 33, lines 743-747).

R1.7) The "Methods" section contains few protocols for the parameter analysis of the imaging data. The authors should describe them. Especially, the definition of diffusion coefficient,

colocalization, and parameters of clustering analysis should be clearly described because they are varied between papers.

We thank Reviewer 1 for this helpful suggestion. While we have published most image analysis methods in detail in previous studies, we agree that it would be helpful to include additional details also here to help the reader. A more thorough description of the key methods has now been added to the "Methods" section (Pages 34-35, lines 758-796).

Reviewer #2

(Remarks to the Author)

This manuscript from Jobin and Calebiro reveals that a class C GPCR, the GABAB receptor, interacts with filamin via its intracellular loop 1 (IL1). This interaction results in a striking striated organisation of the receptor in cells due to tethering to the actin cytoskeleton. Single particle tracking experiments show that GABAB receptor interaction with filamin controls receptor diffusion and that receptor mobility changes following receptor stimulation, which corresponds with a loss of the filamin interaction. Finally, the authors show that the GABAB-filamin interaction prolongs G protein activity following transient stimulation of the GABAB receptor.

This is a beautiful piece of work, with very clear and well-organised figures, and a great narrative. The methodology is clearly described and robust. The authors have used an impressive array of high-resolution approaches to make these observations, and their conclusions clearly show the importance of receptor organisation in efficient cell responses. This is an important study for the GPCR field.

We thank Reviewer 2 for recognizing the importance and robustness of our results and for the helpful suggestions for improvement.

I have a few questions and comments for the authors to address:

R2.1) *Start of results section on pg 10 line 197: "When SNAP-tagged GABAB2 was transfected alone there was no striated pattern". The authors conclude from this that the GABAB2 subunit does not control filamin/actin localisation. However, I can't see that the authors did the reverse experiment to test whether the striated pattern was still present when the GABAB1 subunit was transfected alone.*

There is high homology between the intracellular loops of the two GABAB subunits, and the authors found that replacing the loops of GABAB1 with those of GABAB2 did not change the striated organisation. This seems to suggest to me that these regions of GABAB1 and GABAB2 are interchangeable. In contrast, replacement of the intracellular loops of GABAB1 with those of the mGluR2 removed the patterning.

As currently described, it is not clear to me why the authors conclude that the GABAB2 does not localise to filamin/actin and that this interaction is only mediated by GABAB1. Is it possible that both GABA subunits can control localisation?

Please show whether (or not) transfection of GABAB1 alone results in a striated organisation and discuss these conclusions in more detail. Replacement of the GABAB2 IL1 with that of mGluR2 could also answer this question.

We thank Reviewer 2 for raising these important questions. Unfortunately, the proposed experiment is not feasible because the GABA_{B1} subunit cannot reach the cell surface without the presence of the GABA_{B2} subunit. This is because the GABA_{B1} C-terminal domain contains an ER retention domain that is masked only upon formation of the GABA_{B1}-GABA_{B2} heterodimer, ensuring that only functional heterodimers reach the cell surface. If GABA_{B1} is transfected alone, the ER retention domain prevents its trafficking to the plasma membrane.

Our conclusion that GABA_{B2} does not mediate the actin tethering of the GABA_{B1}-GABA_{B2} was mainly based on the evidence (shown in ED Fig. 1) that GABA_{B2}, when expressed alone, does not show any striated pattern.

Motivated by the Reviewer's suggestion, we have performed a new experiment in which we have generated a GABA_{B1} receptor construct lacking the ER retention domain and expressed it alone in CHO cells. This modified GABA_{B1} is able to reach the cell surface in the absence of GABA_{B2} and, importantly, is arranged in a striated pattern like the GABA_{B1}-GABA_{B2} heterodimer. These new results further demonstrate that the GABA_{B1} subunit mediates GABA_B tethering to the actin cytoskeleton and that the GABA_{B2} subunit is dispensable. These new results have been added to ED Fig. 1d and the corresponding "Results" section (Page 10, lines 217-220).

Extended Data Figure 1d. SNAP-GABA_{B1} deletion mutant lacking the ER retention domain and PDZ motif (ΔER-ΔPDZ) expressed alone. Scale bar, 10 μm.

R2.2) Please show the complete G protein activity time courses for all conditions (at least as supplementary figures) in addition to the inset panels currently displayed as Fig 5c and f, and summarised as estimated time constants in Fig 5 d and g. This will allow for stronger contrast between the effects on G protein activation vs deactivation.

We thank Reviewer 2 for this suggestion. We have now added the complete time course of G protein activity in ED Fig. 6e and 6f.

Extended Data Figure 6e, f. Full time courses of G protein activation and deactivation in HEK293A cells expressing DsRed-FLNA17-18/DsRed-FLNA19-20 (e) or in M2 and A7 cells (f) after transient stimulation with GABA.

R2.3) Can you please comment in the discussion on the effect of transfection of SNAP-GABA subunits into primary hippocampal neurons. Would you hypothesise that endogenous populations be even more restricted in their distribution/location? Based on your experiments, is there a large difference in protein expression for exogenous vs endogenous GABA_B receptors in neurons?

We thank the Reviewer for raising this valid point. Single-particle experiments are performed at very low expression levels, typically less than 0.5 molecules per square micron, to be able to follow individual molecules. These low densities are typically comparable or sometimes below the endogenous levels for most GPCRs. GABA_B densities in hippocampal neurons are relatively high and have been estimated to be in the range of 15-100 particles/μm² (Martín-Belmonte et al., doi: 10.3390/ijms21072459; Booker et al., doi: 10.7554/eLife.51156; Booker et al., doi: 10.1007/s00429-017-1427-x). Therefore, we do not expect the addition of a relatively small fraction of labelled exogenous GABA_B to cause any relevant changes in GABA_B receptor densities while essentially recapitulating the behaviour of the underlying endogenous receptors. We have clarified this point in the manuscript (Page 15, Line 321-323).

R2.4) Cartoon of conclusions (Fig 5h) would be clearer if there was an arrow pointing from the first to the second panel to indicate a sequence of events. I also think FLNA19-20 in the second panel should be replaced with the Filamin A cartoon from the first panel – to show what happens in a “normal cell”. You can then make it clear in the legend that FLNA19-20 can substitute for Filamin A. Consider even labelling the third panel as 5i, to be clear that this is what happens in the absence of filamin (e.g. M2 cells).

We thank the Reviewer for these important suggestions for improvement, which we have followed (see revised Fig. 5h).

Figure 5h. Schematic summary of the results. The interactions of GABA_{B1} with FLNA tethers GABA_B receptors to the actin cytoskeleton, controlling their localization on the plasma membrane. In addition, interaction of either full length FLNA or FLNA19-20 with GABA_{B1} leads to prolonged G_i signaling after transient GABA stimulation, presumably via an allosteric mechanism. This effect is lost in M2 cells lacking FLNA.

Minor comments:

R2.5) Labelling of MCC graphs – please re-label some of the MCC graphs to make it clearer what conditions the co-incident detection quantification is referring to. e.g. the legend in Fig 2h “GABAB1 (vs Actin)” is easier to interpret than the current legends of Fig 2e

We thank the Reviewer for this comment. We have made the suggested modifications to Fig. 1d and Fig. 2b, e.

R2.6) How many independent transfections/SNAP labellings does the number of cells (e.g. Fig 2b legend states n=11) come from? Please also include this information in the figure legend.

We thank the Reviewer for this suggestion, which we have followed in the revised figure legends. As a note, it is worth mentioning that in the single cell experiments presented in this study the variability between individual cells within the same experiment (i.e. same transfection/SNAP labelling) is generally higher than the variability (of cell averages) between different independent experiments.

R2.7) Can you please rationalise the switch to HEK293 cells from CHO cells for FRET experiments examining G protein signalling in the results section.

The choice was dictated by important technical considerations. The single-cell FRET measurements presented in this study require robust cell surface expression of GABA_B heterodimers as well as of the fluorescently tagged G protein subunits. HEK293 are largely superior to CHO cells both in terms of transfection efficiency and cell surface expression, which was required to perform reliable FRET measurements. However, HEK293 as well as most other cell lines do not adhere uniformly to coverslips, which is a major limitation for single-particle tracking experiments as they present patches of the plasma membrane that are not visible in TIRF. CHO cells, in contrast, adhere very well and present a particularly flat surface, which makes them an ideal model for single-particle tracking experiments at the plasma membrane. We have added a comment on that point in the “Results” section of the manuscript (Page 18, line 375).

R2.8) The specificity of the peptide array experiments is quite striking. I was wondering whether you could add the normalised data for these experiments as a supplementary data table, so the reader can get an indication of what is designated a high, medium or low interaction on the heat map.

We thank the Reviewer for this comment. We were also pleased by the specificity of the peptide array measurements. Based on the Reviewer’s suggestion, we have now included the raw intensity data in ED Fig. 4c.

c

	GABA _B IL1	FLNA 19-20	FLNA 17-18	Blank
612	NIYN ⁶¹² SHVRYIQNSQP ⁶³⁰ NLNN	0.81	0.12	0.04
	A ⁶¹² IYN ⁶¹² SHVRYIQNSQP ⁶³⁰ NLNN	1.90	0.27	0.02
	N ⁶¹² A ⁶¹² YN ⁶¹² SHVRYIQNSQP ⁶³⁰ NLNN	0.36	0.12	0.01
	NI ⁶¹² A ⁶¹² NSHVRYIQNSQP ⁶³⁰ NLNN	0.08	0.04	0.01
	NIYA ⁶¹² SHVRYIQNSQP ⁶³⁰ NLNN	1.17	0.07	0.12
	NIYNA ⁶¹² HVRYIQNSQP ⁶³⁰ NLNN	0.15	0.05	0.02
	NIYNSA ⁶¹² VRYIQNSQP ⁶³⁰ NLNN	1.22	0.07	0.02
	NIYNSH ⁶¹² ARYIQNSQP ⁶³⁰ NLNN	0.32	0.05	0.03
	NIYNSHV ⁶¹² RAYIQNSQP ⁶³⁰ NLNN	0.10	0.03	0.04
	NIYNSHVRA ⁶¹² IQNSQP ⁶³⁰ NLNN	0.06	0.04	0.02
	NIYNSHVRYA ⁶¹² QNSQP ⁶³⁰ NLNN	0.14	0.05	0.03
	NIYNSHVRYL ⁶¹² ANSQP ⁶³⁰ NLNN	5.55	0.26	0.05
	NIYNSHVRYIQ ⁶¹² ASQP ⁶³⁰ NLNN	1.42	0.12	0.03
	NIYNSHVRYIQNA ⁶¹² QP ⁶³⁰ NLNN	1.27	0.13	0.04
	NIYNSHVRYIQNSA ⁶¹² P ⁶³⁰ NLNN	1.18	0.12	0.06
	NIYNSHVRYIQNSQ ⁶¹² ANLNN	0.68	0.08	0.03
	NIYNSHVRYIQNSQP ⁶¹² ALNN	1.31	0.13	0.04
	NIYNSHVRYIQNSQP ⁶¹² NANN	0.93	0.19	0.04
	NIYNSHVRYIQNSQP ⁶¹² NLAN	1.16	0.23	0.02
	NIYNSHVRYIQNSQP ⁶¹² NLNA	0.98	0.13	0.03

Extended Data Figure 4c. Intensity values of the peptide microarray analysis shown in Fig. 3.

Reviewer #3

(Remarks to the Author)

The actin cross-linking and scaffolding protein filamin has been implicated in connecting a range of membrane proteins to the cytoskeleton. Here, Jobin et al., report that they have extended that list to include GABA-B. While the report contains some potentially interesting observations, evidence for the central conclusion – that GABA-B binds full-length filamin – is weak. Beyond co-localization, no data on GABA-filamin protein-protein interactions are provided. The binding site on FLNA is assumed to be FLNA9-20 although other sites were not examined and other than a peptide array FLNA19-20 is not shown to bind GABA. The filamin-GABA interaction is central to the argument being made and needs to be much more rigorously established. The functional relevance of the interaction is also poorly developed and appears to be modest at best. The authors begin focusing on GABA localization but, if I understand correctly, they conclude that rather than being importance for localizing the protein, filamin binding is important in regulating GABA signaling. The authors should draw more on their point mutated GABA to support their conclusions and I suggest that FLNA knockout or knockdown be employed to a greater extend to validate conclusions.

We thank Reviewer 3 for carefully reviewing our manuscript and for their suggestions for improvement. Based on the Reviewers' suggestions we have performed a series of new experiments and analyses, including additional controls (e.g. CD86 as "irrelevant membrane protein" control), which provide additional evidence to support our main conclusion.

Regarding the functional relevance of GABA_B-FLNA interactions, the Reviewer raises a series of valid points, which we feel we had not sufficiently discussed and clarified in our original submission (see also our response to Reviewer 1). Based on our findings, we propose that there are at least two separate mechanisms through which GABA_B-FLNA interactions impact on GABA_B function. A first important mechanism is via controlling GABA_B precise location by tethering it to the subcortical actin cytoskeleton. It is in fact well established that synaptic transmission requires a highly organized spatial organization of receptors at synapses and that this is typically achieved via (scaffold-mediated) interactions with the underlying cytoskeleton. Therefore, it appears highly plausible that the FLNA-GABA_B interactions uncovered in our study participate in correctly positioning GABA_B receptors at synapses, and, thus, indirectly impact on GABA_B signalling. The second mechanism involves an additional "allosteric" effect that we have uncovered whereby the interaction with FLNA appears to also more directly fine-tune G protein signalling downstream of GABA_B. The two mechanisms are distinct and likely cooperate to ensure a precise spatiotemporal control of GABA_B signalling. These important points are now clarified in the Discussion (Page 22, Lines 466-478).

Regarding the conclusion that GABA_B binds full-length FLNA, the Reviewer's view that the evidence is weak appears not to be shared by the other Reviewers and we would also respectfully disagree. In fact, we provide several independent pieces of evidence that have been further strengthened by additional experiments and controls included in the revision (see responses to specific Reviewers' points for details). Altogether, we believe that our revised manuscript provides strong evidence to support the conclusion that GABA_B-FLNA interactions dynamically tether GABA_B to the actin cytoskeleton. Our key independent findings include:

1) In vitro peptide microarray binding data showing that the FLNA19-20 repeats bind with high specificity to the IL1 of GABA_{B1}.

2) Single molecule evidence in living cells show that GABA_B and full length FLNA undergo dynamic interactions above what would be expected for random colocalizations (or seen with a FLNA binding GABA_B mutant).

3) The observation that FLNA19-20 domains when expressed in cells have a potent dominant negative effect, virtually abolishing GABA_B localisation on actin fibres, with MCC levels comparable to those observed with the CD86 control. Of note, this dominant negative approach has been employed in previous studies to demonstrate a role of FLNA in tethering membrane receptors to the actin cytoskeleton (see e.g. Peverelli et al., doi: 10.1210/en.2014-1063; Treppiedi et al., doi: 10.1210/en.2018-00368; Najib et al., doi: 10.1128/mcb.06252-11; Lin et al., doi: 10.1159/000065531).

4) The rather remarkable observation that mutating the IL1 FLNA-binding domain in GABA_{B1} identified in 1) (but not any other parts of GABA_{B1/B2}) also abolishes the colocalization of GABA_B with both actin and filamin.

5) The evidence in M2 (lacking FLNA) and A7 (M2 clone in which FLNA was re-expressed), where we show that the typical GABA_B striated pattern along actin fibres is only observed in A7 cells. Despite some potential limitations raised by the Reviewer, it is worth mentioning that these cells have been used in numerous previous studies to investigate FLNA and demonstrate a role of FLNA in tethering membrane proteins to the actin cytoskeleton (see e.g. Taneja et al., doi: 10.1016/j.celrep.2020.03.041; Pons et al., doi: 10.1242/jcs.193821; Onoprishvili et al., doi: 10.1007/s11064-008-9684-y).

6) Finally, new evidence based on FLNA siRNA silencing, as suggested by the Reviewer, which provides additional independent evidence that FLNA knockdown disrupts the GABA_B striated arrangement on the plasma membrane (see below for details).

Regarding the FLNA domain mediating GABA_B interactions, FLNA19-20 domains have been previously shown to be key domains mediating the interactions with other GPCRs (see e.g. Treppiedi et al., doi: 10.1210/en.2018-00368 ; Peverelli et al., doi: 10.1210/en.2014-1063 ; 2014, Endocrinology, Najib et al., doi: 10.1128/mcb.06252-11; Enz et al., doi: 10.1042/BJ20021750; reviewed in Tirupula et al., doi: 10.1021/acs.biochem.5b00975). Whereas our findings *in vitro* and in cells suggest that FLNA19-20 domains, but not FLNA17-18, bind GABA_{B1} IL1 and participate in tethering GABA_B to the actin cytoskeleton, we agree with the Reviewer that we cannot rule out the additional involvement of other FLNA repeats. Whereas it would be interesting to systematically investigate the role of individual FLNA repeats in the future, this would not change in any way our major conclusion that FLNA mediates GABA_B tethering to the actin cytoskeleton and goes beyond the scope of our present study, which was to identify the involved scaffold and study its impact on the spatiotemporal organisation of GABA_B receptors at the plasma membrane. A paragraph clarifying these important points and the limitations of our study re. the FLNA repeats mediating the interaction with GABA_B has been added to the Discussion (see Page 22, lines 457-465).

Specific comments:

R3.1) *Building on their prior results, the authors show a remarkably strong co-localization of SNAP-GABA-B1 with F-actin (Life-act) in CHO cells (Fig 1a-c). However, they do not include a negative control for this experiment, only showing a calculated homogenous distribution. The data should be validated using an irrelevant SNAP-tagged membrane protein. The dynamic colocalization analysis (Fig 1d,e) indicates that control proteins (CD86) may not behave like a model homogeneous distribution highlighting the need for experimental controls.*

We thank Reviewer 3 for raising these valid and important points. Due to the complex organization of the plasma membrane, no real membrane protein shows perfect homogenous distribution. That is why it is indeed important to include “irrelevant membrane protein” controls in such experiments as suggested by the Reviewer. CD86 is a well-established control previously used in several single-molecule studies on GPCRs (see e.g. Sungkaworn et al., doi: 10.1038/nature24264; Calebiro et al., doi: 10.1073/pnas.1205798110, Dorsch et al., doi: 10.1016/j.conb.2011.12.006) and a very helpful “irrelevant membrane protein” control in our experiments, as it has similar diffusion properties to untethered GABA_B receptors and, at the same time, is well accepted not to have relevant interactions with actin, GPCRs or G proteins.

Based on the Reviewer’s suggestion, we have performed additional analyses in CHO cells co-transfected with SNAP-CD86 and Lifeact-GFP. The results show that the strong co-localization between SNAP-GABA (MCC 0.77) is much higher than expected for an “irrelevant membrane protein” used as control (MCC 0.34 for CD86; 0.15 for homogeneous distribution). We have added the new results in Fig. 1c and in the corresponding “results” section (see Page 5, lines 99-104).

In addition, it is worth noting that, contrary to what one might perhaps wrongly assume, MCCs are often substantially higher than 0 even in the theoretical limit of perfect homogenous distributions (see e.g. Fig. 1c and following). As this can be misleading to both experts and non-experts, we believe it is also important to show the expected MCC values for perfect homogenous distributions.

R3.2) *In seeking potential linkers to the cytoskeleton the authors tested FLNA. FLNA colocalizes with F-actin so, as the authors have already shown GABA co-localization with F-actin, it is unsurprising that FLNA and GABA also colocalize. This co-localization in transfected cells or neurons need not implicate FLNA in the link. Again, a negative control SNAP tagged protein is missing.*

We thank Reviewer 3 for raising this important point and for the suggestion. In our colocalization experiments, we have already included an ideal negative control, i.e. the SNAP-tagged mutant GABA_{B1} subunit in which the identified FLNA-binding region, i.e. GABA_{B1} IL1, was replaced with the IL1 of mGluR2 (see Fig. 3a,b). Importantly, the results show a robust loss of colocalization with both F-actin and FLNA in the case of the GABA_{B1} IL1 mutant (MCC decrease from ~0.8 to ~0.4). In addition, we have now included additional experiments and analyses of the colocalization between SNAP-CD86, used as “irrelevant membrane protein” control, and both Lifeact-GFP and GFP-FLNA, as suggested by the Reviewer. Of note, the remaining MCC of ~0.4 observed between the GABA_{B1} IL1 mutant and FLNA is comparable to the values observed between CD86 and actin or FLNA, suggesting that GABA_B interaction with F-actin/FLNA is virtually abrogated by removal of the GABA_B FLNA binding site in IL1.

We definitely agree with the Reviewer that because FLNA colocalizes with F-actin, GABA_B colocalization with FLNA does not per se prove a link. The link is provided by the peptide microarray results (Fig. 3 and ED Fig. 4), single-molecule evidence that GABA_B and full length FLNA undergo dynamic (and GABA-modulated) interactions and co-trapping above what would be expected for random colocalizations or observed with the GABA_{B1} IL1 mutant (Fig. 4d-g), the results with the FLNA19-20 dominant negative fragment, the results comparing M2 and A7 cells, as well as the results of new FLNA knockdown experiments with siRNAs (see also our response to the Reviewer’s general comments).

We have better clarified these important points in the discussion (see Page 22, lines 457-465). The new results with CD86 have been added in Fig. 1c and 2b.

R3.3) *The authors attempt to further implicate FLNA by expressing a small fragment of FLNA19-20 with the idea that it might act as a dominant negative fragment. This strategy is difficult to justify – FLNA is reported to bind many proteins and while binding sites do seem to be somewhat enriched in the rod 2 region, selecting only 2 of the total 24 repeats from filamin, especially in the absence of clear data that GABA binds that region, is questionable. Likewise, the statement that that FLNA17-18 are not involved in membrane protein interaction is incorrect (e.g. PMID 16293600, but many other partners may engage FLNA17). Why have the authors not reported the impact of loss of FLNA on GABA localization?*

We do agree with the Reviewer that FLNA contains multiple Ig repeats, several of which have been shown to interact with a number of proteins. Most FLNA-interacting proteins appear to bind repeats 17 to 24 (Popowicz et al., doi: 10.1016/j.tibs.2006.05.006). Previous studies on other GPCRs have already demonstrated that FLNA can directly interact with several receptors such as D2, D3, AT1, MAS and SST2 (Lin et al., doi: 10.1073/pnas.011538198; Tirupula et al., doi: 10.1021/acs.biochem.5b00975, Trepieddi et al., doi: 10.1210/en.2018-00368 ; Peverelli et al., doi: 10.1210/en.2014-1063 ; 2014, Endocrinology, Najib et al., doi: 10.1128/mcb.06252-11). In the case of the SST2, a series of experiments, including in vitro SPR with repeats 17 to 24, has mapped the direct, high-affinity binding site to FLNA19-20 (Najib et al., doi: 10.1128/mcb.06252-11, Trepieddi et al., doi: 10.1210/en.2018-00368).

Importantly, the use of a FLNA19-20 fragment (and FLNA17-18 as a control) to interfere with FLNA binding to GPCRs has been previously successfully used to demonstrate FLNA interaction with the somatostatin SST2 receptor both in vitro with purified proteins (Najib et al doi: 10.1128/mcb.06252-11) and in living cells (Peverelli et al., doi: 10.1210/en.2014-1063). This also includes a previous single-molecule microscopy study from our group (Trepieddi et al, doi: 10.1210/en.2018-00368), where we successfully utilised FLNA19-20 expression (with FLNA17-18 as control) to dominant-negatively disrupt SST2/FLNA interactions in living cells.

Given the overall high conservation among GPCRs, we hypothesized that a similar mechanism might also occur in the case of GABA_B, which is what we set out to investigate. Remarkably, the degree of tethering to the actin cytoskeleton and the displacement caused by FLNA19-20 is much stronger in the case of GABA_B than SST2 or other GPCRs we have tested so far, suggesting that this phenomenon is particularly relevant in the case of GABA_B.

Although the FLNA17-18 fragment does indeed bind other membrane proteins, our data, including our peptide microarray results, clearly shown that it does not interact with GABA_B, making it an ideal negative control in our study. We have clarified these relevant points in the revised manuscript (see ED Fig. 4).

Importantly, a key advantage of using a transient dominant negative approach with FLNA fragments is that it does not impact on the actin cross linking function of FLNA. In contrast, chronic knockout or knockdown of FLNA is expected to affect the organization of the actin cytoskeleton, and could therefore alter the spatial organization of GABA_B receptors irrespectively of whether they directly interact or not with FLNA. This makes our dominant negative approach with FLNA fragments more suitable to selectively investigate the consequences of GABA_B-FLNA interactions on the spatiotemporal organization of GABA_B receptors at the plasma membrane.

Together with the already mentioned complementary approaches that we have used in this study (including mutation of the IL1 FLNA binding site, experiments in M2 cells lacking FLNA, etc.), we believe that our results with FLNA fragments provide convincing evidence that FLNA plays a key role in GABA_B tethering to the acting cytoskeleton.

At the same time, we do agree that other FLNA domains might also play a role. Whereas a systematic investigation of all the FLNA repeats that might be involved in GABA_B binding goes beyond the scope of our present study, this is certainly an important point that we have now carefully discussed and clarified in the revised manuscript (see Page 22, lines 457-465).

R3.4) *In the images shown in Fig 2d, GABA-B1 seems to be totally mislocalized by the over-expressed FLNA19-20, however quantification results in an average MCC of ~0.5. Is the image shown representative? This experiment also requires additional controls, e.g., does FLNA19-20 alter the localization of other actin-binding membrane proteins?*

We thank Reviewer 3 for these important comment. The image shown in Fig. 2d is representative (MCC 0.59) and additional representative examples of TIRF images obtained in this condition, i.e. when GABA_{B1} is co-transfected with GABA_{B2} and FLNA19-20, are provided below.

(Images shown for reviewing purposes)

One might wonder why such a large mislocalization from actin fibres still gives MCC ~0.5. As explained in the response to **R3.1**, this is largely due to the nature of MCCs. While MCCs are robust and widely used, the interpretation of their values is complicated by two factors. First, the MCC does not necessarily give a value of 0 even in the case when one of the two channels has a perfect homogenous distribution. This is why we are showing the expected values for homogenous distributions to highlight what would be the lowest theoretically possible values. Second, no membrane protein has a perfect homogenous distribution. For this reason, we have added an “irrelevant membrane protein” control, i.e. CD86, which has no strong interactions with the actin cytoskeleton, as suggested by the Reviewer. As the MCC values obtained in the case of CD86 are ~0.4, the effect of FLNA19-20 corresponds indeed to a large mislocalization, consistent with what can be seen looking at the images. A sentence clarifying these considerations about the interpretation of MCCs has been added to the manuscript (see Page 5, lines 98-99).

Regarding the question whether FLNA19-20 could also alter the localization of other membrane proteins, the answer is yes. Since FLNA19-20 is known to mediate the interaction of FLNA with other membrane proteins, we anticipate that expressing FLNA19-20 could also specifically modify the localization of those proteins. We have actually already shown that this is the case for SST2, which is also partially anchored to the actin cytoskeleton via FLNA (Peverelli et al., doi: 10.1210/en.2014-1063; Treppiedi et al, doi: 10.1210/en.2018-00368). Whereas FLNA19-20 could indeed be used as a tool to investigate other FLNA-interacting membrane proteins as well, performing those experiments goes beyond the scope of our study and their results would not change our conclusions.

R3.5) *Using chimeric receptors, the authors convincingly show that the intracellular loop 1 is important for co-localization of GABA-B1 with actin. However, the conclusion that the B2 receptor is not involved (extended data) is compromised by its expression in the absence of*

B1. Does B1 expressed alone target to actin? Does mutagenesis of the B2 loop – in the context of the heterodimer – impact localization?

We thank Reviewer 3 for raising this relevant point, which requires a clarification. The GABA_{B1} subunit cannot reach the cell surface alone, i.e. without the presence of the GABA_{B2} subunit (see also our response to Reviewer 2, **R2.1**). This is because the GABA_{B1} C-tail contains an ER retention motif that is masked only upon formation of the GABA_{B1}-GABA_{B2} heterodimer (see Introduction Page 3, lines 58-60). This is part of a key physiological mechanism to ensure that only functional heterodimers reach the cell surface. If GABA_{B1} is transfected alone, the ER retention domain prevents its trafficking to the plasma membrane.

Based on the Reviewer's suggestion, we have generated a GABA_{B1} subunit construct lacking the ER retention domain and expressed it alone in CHO cells. Importantly, this modified GABA_{B1} is able to reach the cell surface in the absence of GABA_{B2} where it shows a striated pattern like the GABA_{B1}-GABA_{B2} heterodimer. These new results further demonstrate that the GABA_{B1} subunit mediates GABA_B tethering to the actin cytoskeleton and that the GABA_{B2} subunit is dispensable. The results have been added to ED Fig. 1d and the corresponding "Results" section (Page 10, lines 217-220).

R3.6) *To show association with filamin the authors turned to peptide arrays and the FLNA19-20 fragment. Specificity of the peptide arrays is based in part on comparison of the FLNA19-20 and FLNA17-18 proteins but no validation that these proteins (expressed as DsRed fusions) were stably expressed at the expected sizes and at comparable levels was provided.*

We thank Reviewer 3 for suggesting the inclusion of this important control. Based on the Reviewer's suggestion, we have performed a western blot analysis on extracts from cells expressing either FLNA fragment. The results, which has been added to ED Fig. 4b, confirm the correct expression of both FLNA fragments at the expected size. Despite the control FLNA17-18 being expressed at moderately higher levels, FLNA-19-20 showed much stronger binding than FLNA17-18 to the IL1 of GABA_{B1} (Fig. 3). To allow readers to perform a more quantitative comparison, we have also added the raw intensity data of the peptide microarray measurements in ED Fig. 4c.

R3.7) *Using the peptide array, point mutations in GABA-B1 that impact binding were detected but these potentially useful mutations were not employed in any functional experiments.*

We agree with the Reviewer that some of the identified point mutations in GABA_{B1} could be useful in functional experiments. However, the detailed peptide microarray results became available only late in our study, which prevented their further use for functional studies. Since we have already performed a thorough mutagenesis study based on systematic domain deletion and swapping with mGluR2 (ED Fig. 1) and subsequent investigations of the identified IL1 mutant, in the revision we have decided to prioritize other important experiments suggested by the Reviewers.

R3.8) *The use of the M2 and A7 cell lines supports a role for filamin but these lines are problematic in many ways. Notably, they were generated a long time ago and have accumulated differences in protein expression over time making, comparison of phenotypes difficult. The use of knockout cells (or knockdown cells) acutely reconstituted with FLNA or FLNA lacking the 19-20 would provide a more robust approach.*

We thank the Reviewer for acknowledging that our result in M2 and A7 cell lines support a role for FLNA in tethering GABA_B to the actin cytoskeleton.

Despite the fact that M2/A7 cells were generated a number of years ago and the potential for divergence over time mentioned by the Reviewer, M2 and A7 continue to be a widely used and generally accepted model – see for instance several recent publications on FLNA (Taneja et al., doi: 10.1016/j.celrep.2020.03.041; Cho et al., doi: 10.1074/jbc.M115.638445; Guo et al., doi: 10.1080/22221751.2019.1632153).

As already mentioned in our response to **R3.3**, there are also some additional limitations of using FLNA KO cells or FLNA silencing besides the specific points about M2/A7 cells raised by the Reviewer. Since FLNA plays key roles in actin crosslinking and network dynamics, FLNA KO or silencing is expected not only to disrupt GABA_B actin tethering but also to affect the overall organization of the actin cytoskeleton, potentially altering the spatial organization of GABA_B receptors irrespective of whether they directly interact or not with FLNA. Moreover, FLNA and the actin cytoskeleton are implicated in many essential cell functions such as adhesion, replication, migration, etc, and, thus, FLNA KO/silencing could have other indirect effects on GABA_B receptors. These limitations warrant against relying on FLNA KO/silencing as the only (or main) approach to investigate the role of FLNA-GABA_B interactions in living cells.

Despite these disadvantages, we have followed the Reviewer's suggestion and performed FLNA siRNA silencing experiments in CHO cells (see below and ED Fig. 2). The results indicated that FLNA silencing abolishes the typical GABA_B striated pattern on the plasma membrane, proving yet another independent piece of evidence to support our main conclusion. The new results are now shown in ED Fig. 2. Additional representative images of cells transfected with FLNA siRNAs or scrambled siRNA used as control are provided below.

Extended Data Figure 2a. Western blot analysis showing the efficiency of FLNA knockdown in CHO cells. Lysates of untransfected cells or cells transfected with scrambled siRNA or two separate siRNAs (#1 and 2) against FLNA were probed with an anti-FLNA antibody. GAPDH was used as a loading control.

siRNA scrambled

siRNA FLNA KO #1 + #2

(Images shown for reviewing purpose; scale bars 10 μ m)

Motivated by the Reviewer's suggestions, we have also actually attempted to generate our own FLNA CRISPR KO CHO cells as well as asked a company to generate FLNA CRISPR KO HEK cells. Perhaps not surprisingly, all those attempts have repeatedly failed, which we strongly suspect is due to detrimental effects of FLNA KO on cell survival/replication. This may explain why despite M2/A7 having been derived several years ago, no superior cell tools have been generated in the meantime.

Altogether, we believe that our results obtained with independent and complementary approaches, which include peptide microarrays, identification and replacement of the GABA_{B1} IL1 FLNA binding site, use of dominant negative FLNA19-20 fragment to displace GABA_B from FLNA in living cells, single particle tracking experiments investigating individual GABA_B-FLNA interactions, the use of FLNA KO siRNA and the comparison between M2 and A7 cells, which all have specific strengths and limitations when considered in isolation, provide rather strong evidence to support the conclusion that GABA_B-FLNA interactions dynamically tether GABA_B to the actin cytoskeleton.

REVIEWERS' COMMENTS

Reviewer #1 (Remarks to the Author):

questions.

1. Fig. 4g:

What are the unit and normalization?

Probably, the numbers mean the numbers of GABA (or FLNA?) molecules colocalized with FLNA (or GABA?) per unit area. Then, what does “per cell” mean?

The colocalization probability must depend on the densities of GABA and FLNA molecules. It is unclear how the colocalization probabilities were normalized to the molecular densities.

2. Discussion on the functions of GABA/FLNA association:

The authors discussed the possibilities of the positive effect of GABA/FLNA colocalization on GABA signaling. Although such possibilities cannot be excluded, the results in this study indicate dissociation of GABA and FLNA upon GABA activation, which may indicate that FLNA holds GABA in the inactive form and the active form of GABA does not interact with FLNA. A more precise discussion on this (apparent?) discrepancy is desired.

Reviewer #2 (Remarks to the Author):

This is a very elegant and exciting study. The authors have addressed all my comments and I recommend publication in Nature Communications.

Reviewer #3 (Remarks to the Author):

The authors have revised their manuscript in response to reviewer comments – this has clarified/resolved several important points.

While I still feel that the manuscript would benefit from greater characterization of the filamin-GABA-B interaction and from the use of mutants, rather than dominant negative constructs, to perturb the interaction, I appreciate the large amount of work already present and the others reviewers' positive evaluation of the manuscript.

Minor points:

Rather than relying solely on the dominant-negative construct the authors now include supplementary figures showing loss of the striated GABA-B pattern upon siRNA silencing of FLNA. Having generated these data why did the authors not stain for F-actin and calculate co-localization in the FLNa knockdown cells to directly compare knockdown and inhibitor studies (Extended data fig 2)? This would have strengthened these data.

In their response to reviewers the authors note their difficulty in generating FLNA knockout cells – I find this surprising considering the reports of filamin knockout and knockdown cells in the literature, but it may reflect the specific cell lines used. They should consider noting this in the manuscript to explain why they did not undertake more knockout studies.

Response to reviewers

Reviewer #1 (R1)

We thank the reviewer for carefully reading our manuscript and for their suggestions for improvement.

(Remarks to the Author):
questions.

R1.1) Fig. 4g: What are the unit and normalization?

Probably, the numbers mean the numbers of GABA (or FLNA?) molecules colocalized with FLNA (or GABA?) per unit area. Then, what does “per cell” mean?

The colocalization probability must depend on the densities of GABA and FLNA molecules. It is unclear how the colocalization probabilities were normalized to the molecular densities.

The y-axis represents densities, i.e. the number of GABA_B/FLNA colocalizations in each diffusivity state divided by the cell surface and averaged for the first 200 frames. The use of “per cell” was incorrect and we have modified the y-axis label accordingly.

R1.2) Discussion on the functions of GABA/FLNA association:

The authors discussed the possibilities of the positive effect of GABA/FLNA colocalization on GABA signaling. Although such possibilities cannot be excluded, the results in this study indicate dissociation of GABA and FLNA upon GABA activation, which may indicate that FLNA holds GABA in the inactive form and the active form of GABA does not interact with FLNA. A more precise discussion on this (apparent?) discrepancy is desired.

We agree with the Reviewer that the interplay between FLNA, GABA_B and the actin cytoskeleton is complex and future studies will be required to fully understand its functional consequences. Our data suggest that binding of FLNA19-20 to GABA_B prolongs GABA signalling. At the same time, agonist stimulation weakens GABA_B interaction with FLNA causing its partial redistribution away from actin fibres. This could have negative as well as positive consequences on signalling, for instance by promoting GABA_B internalization and signalling at intracellular sites, which has now been shown for several GPCRs. In the revised manuscript, we have added a sentence to briefly discuss these points (Page 14, Lines 342-343)

Reviewer #2 (R2) (Remarks to the Author):

This is a very elegant and exciting study. The authors have addressed all my comments and I recommend publication in Nature Communications.

We thank the reviewer for their positive comments and valuable suggestions to improve our study.

Reviewer #3 (R3) (Remarks to the Author):

The authors have revised their manuscript in response to reviewer comments – this has clarified/resolved several important points.

While I still feel that the manuscript would benefit from greater characterization of the filamin-GABA-B interaction and from the use of mutants, rather than dominant negative constructs, to perturb the interaction, I appreciate the large amount of work already present and the others reviewers' positive evaluation of the manuscript.

We thank the reviewer for their thoughtful and constructive comments and for recognizing the large amount of work present in our study.

Minor points:

R3.1) *Rather than relying solely on the dominant-negative construct the authors now include supplementary figures showing loss of the striated GABA-B pattern upon siRNA silencing of FLNA. Having generated these data why did the authors not stain for F-actin and calculate co-localization in the FLNA knockdown cells to directly compare knockdown and inhibitor studies (Extended data fig 2)? This would have strengthened these data.*

FLNA silencing in CHO cells caused a very clear virtual disappearance of the typical GABA_B striated pattern, which strongly indicates a loss of the normal actin localization. Since visualizing actin could have caused artefacts due to the requirement for a second transfection or cell fixation/staining, we opted not to include actin staining in these supplementary experiments.

R3.2) *In their response to reviewers the authors note their difficulty in generating FLNA knockout cells – I find this surprising considering the reports of filamin knockout and knockdown cells in the literature, but it may reflect the specific cell lines used. They should consider noting this in the manuscript to explain why they did not undertake more knockout studies.*

We agree with the reviewer that the difficulties in generating two different FLNA KO cells were not entirely expected and may reflect a particular relevance of FLNA in the two used cell models. As suggested by the Reviewer, we have added a sentence in the manuscript clarifying this aspect (Page 7, Lines 140-142).